# A prion accelerates proliferation at the expense of lifespan

David M Garcia[1,2]*[†], Edgar A Campbell[1†], Christopher M Jakobson[1], Mitsuhiro Tsuchiya[3], Ethan A Shaw[2], Acadia L DiNardo[2], Matt Kaeberlein[3], Daniel F Jarosz[1,4]*

[1]Department of Chemical & Systems Biology, Stanford University School of Medicine, Stanford, United States; [2]Institute of Molecular Biology, Department of Biology, University of Oregon, Eugene, United States; [3]Department of Pathology, University of Washington, Seattle, United States; [4]Department of Developmental Biology, Stanford University School of Medicine, Stanford, United States

**Abstract** In fluctuating environments, switching between different growth strategies, such as those affecting cell size and proliferation, can be advantageous to an organism. Trade-offs arise, however. Mechanisms that aberrantly increase cell size or proliferation—such as mutations or chemicals that interfere with growth regulatory pathways—can also shorten lifespan. Here we report a natural example of how the interplay between growth and lifespan can be epigenetically controlled. We find that a highly conserved RNA-modifying enzyme, the pseudouridine synthase Pus4/TruB, can act as a prion, endowing yeast with greater proliferation rates at the cost of a shortened lifespan. Cells harboring the prion grow larger and exhibit altered protein synthesis. This epigenetic state, [*BIG*+] (*better in* growth), allows cells to heritably yet reversibly alter their translational program, leading to the differential synthesis of dozens of proteins, including many that regulate proliferation and aging. Our data reveal a new role for prion-based control of an RNA-modifying enzyme in driving heritable epigenetic states that transform cell growth and survival.

*For correspondence:
dmgarcia@uoregon.edu (DMG);
jarosz@stanford.edu (DFJ)

[†]Equal contributions

## Introduction

Cell size and proliferation are fundamental determinants of development, survival, and disease (*Su and O'Farrell, 1998*). Although they can be independently controlled—for example, when cell division is disrupted, cell size continues to increase, whereas restricting cell growth does not completely inhibit division (*Johnston et al., 1977*)—coupling is common due to their dependence on the same biochemical building blocks (*Su and O'Farrell, 1998*; *Turner et al., 2012*). Changing cell size and proliferation genetically or through chemical perturbation—whether via diet, nutrient sensing (e.g., TOR signaling), or growth factor signaling and stress responses—also impacts lifespan across eukaryotes (*Fontana et al., 2010*; *Kenyon, 2010*; *Pitt and Kaeberlein, 2015*). The importance of tightly controlling each of these properties is underscored by the many mutations affecting cell size and proliferation that are pathogenic, leading to cancer, developmental abnormalities, and numerous diseases of age (reviewed in *Hanahan and Weinberg, 2011*; *Saxton and Sabatini, 2017*).

Organisms commonly alter the relationship between cell size and proliferation during development (*Su and O'Farrell, 1998*). In oocytes of *Drosophila melanogaster*, for example, a massive expansion in cytoplasmic volume without cellular division precedes a later step of numerous cellular divisions without cytoplasmic growth. The rates of cell size increase and proliferation can be strongly coupled to nutrient sensing and changes in metabolism (*Efeyan et al., 2015*; *Turner et al., 2012*). Thus, organisms can use genetically encoded signaling pathways to commit to different strategies depending on their needs and environment (*Ivanov et al., 2015*; *Jung et al., 2018*). Epigenetic tuning of cell

**eLife digest** Cells make different proteins to perform different tasks. Each protein is a long chain of building blocks called amino acids that must fold into a particular shape before it can be useful. Some proteins can fold in more than one way, a normal form and a 'prion' form. Prions are unusual in that they can force normally folded proteins with the same amino acid sequence as them to refold into new prions. This means that a single prion can make many more that are inherited when cells divide. Some prions can cause disease, but others may be beneficial.

Pus4 is a yeast protein that is typically involved in modifying ribonucleic acids, molecules that help translate genetic information into new proteins. Sometimes Pus4 can adopt a beneficial prion conformation called [$BIG^+$]. When yeast cells have access to plenty of nutrients, [$BIG^+$] helps them grow faster and larger, but this comes at the cost of a shorter lifespan.

Garcia, Campbell et al. combined computational modeling and experiments in baker's yeast (*Saccharomyces cerevisiae*) to investigate the role of [$BIG^+$]. They found that the prion accelerated protein production, leading to both faster growth and a shorter lifespan in these cells, even without any changes in gene sequence.

Garcia, Campbell et al.'s findings explain the beneficial activity of prion proteins in baker's yeast cells. The results also describe how cells balance a tradeoff between growth and lifespan without any changes in the genome. This helps to highlight that genetics do not always explain the behaviors of cells, and further methods are needed to better understand cell biology.

size and proliferation could in principle provide a stable yet reversible mechanism to alter the relationship between these properties according to differing needs in fluctuating environments. Histone modifications can enable such adaptation in a way that can be heritable over several mitotic divisions. However, apart from a few notable exceptions (*Catania et al., 2020*; *Grewal and Klar, 1996*; *Nakayama et al., 2000*), most studied examples are erased during meiosis (*Heard and Martienssen, 2014*; *Moazed, 2011*).

Prions are distinct among epigenetic mechanisms in that they are faithfully transmitted through both mitotic and meiotic cell divisions (*Brown and Lindquist, 2009*; *Cox, 1965*; *Garcia and Jarosz, 2014*; *Wickner, 1994*). The unusual folding landscape of prion proteins, which allows the recruitment of proteins from the naïve to the prion fold, promotes a mode of inheritance that is both stable and reversible (*Chakrabortee et al., 2016a*; *McKinley et al., 1983*). For example, transient perturbations in molecular chaperone activity (*Brown and Lindquist, 2009*; *Chernoff et al., 1995*), specific environmental stressors (*Garcia et al., 2016*; *Singh et al., 1979*; *Tuite et al., 1981*), or regulated proteolysis (*Ali et al., 2014*; *Kabani et al., 2014*) can induce or eliminate prion states. It is now appreciated that this form of information transfer is far more common than previously realized, occurring not only in mammals but also in wild fungi, plants, and even bacteria (*Chakrabortee et al., 2016b*; *Halfmann et al., 2012*; *Yuan and Hochschild, 2017*).

Here we investigate the inheritance of a prion-based epigenetic state that alters yeast cell physiology, potentiating a trade-off between proliferation and lifespan. We first describe the [$BIG^+$] prion—driven by the conserved pseudouridine synthase Pus4 (known as TruB in mammals and bacteria)—which increases proliferation but shortens lifespan. We then quantitatively model the adaptive value of this 'live fast, die young' growth strategy in fluctuating environments. [$BIG^+$] cells are larger and exhibit increased protein synthesis while maintaining Pus4-dependent pseudouridylation activity. Finally, we find evidence for analogous Pus4-dependent epigenetic control of cell size in wild yeast. The epigenetic inheritance of an altered form of an RNA-modifying enzyme over long biological timescales, as occurs in [$BIG^+$], reveals a new mechanism by which regulation of these activities can be perpetuated across generations.

## Results
### Cells bearing a prion-like epigenetic element live fast and die young
We recently discovered more than 40 protein-based epigenetic elements in *Saccharomyces cerevisiae* that are both heritable and reversible upon transient perturbation of protein chaperone

function (*Chakrabortee et al., 2016a*). Many of the proteins that underlie this behavior have the potential to regulate growth. One of these epigenetic states was induced by transient overexpression of the highly conserved pseudouridine synthase *PUS4*/TRUB, which catalyzes the formation of a ubiquitous pseudouridine on U55 in tRNAs in bacteria, yeast, and humans (*Becker et al., 1997*; *Gutgsell et al., 2000*; *Zucchini et al., 2003*). Mutation of U55 leads to large fitness deficits, second only to mutations in the tRNA anticodon loop, highlighting the functional importance of this nucleotide (*Li et al., 2016*). Mutation of U55 is also linked to deafness and diabetes in humans (*Wang et al., 2016*). We originally discovered this epigenetic state because transient overexpression of *PUS4* led to an enduring and heritable growth improvement in medium containing zinc sulfate (*Chakrabortee et al., 2016a*). This phenotype could possibly be explained by the strong genetic interaction between *PUS4* and the zinc homeostasis regulator *ZRC1*—among all genes, *ZRC1* is the second strongest genetic interactor with *PUS4* (*Koh et al., 2010*). Upon closer examination, we noticed that cells harboring the Pus4-induced element also achieved an ~60% faster maximal proliferation rate than naïve cells in the absence of zinc stress in standard rich medium (YPD; *Figure 1A*, p=0.0095, unpaired t-test). We therefore initially named this mitotically heritable element 'Big[+]' for *b*etter *i*n *g*rowth.

We next tested whether this growth advantage from the Big[+] epigenetic element was more pronounced during direct competition and oscillating nutrient availability—as a single-celled organism such as yeast often faces in nature. To do so, we performed a competition experiment that encompassed periods of abundant nutrient availability followed by starvation. Using resistance to canavanine as a marker to distinguish naïve and Big[+] strains, we mixed equal numbers of cells from each strain. We propagated the mixed culture for approximately 100 generations, diluting into fresh medium and measuring the fraction of the population that harbored the canavanine resistance marker every 10 generations (*Figure 1—figure supplement 1A*). In these experiments, cells harboring Big[+] invariably outcompeted the genetically identical naïve cells that lacked it—they *live fast*—with a selection coefficient of nearly 1% (*Figure 1B*). Competitions using reciprocally marked strains produced equivalent results. As a frame of reference, the fitness advantage that we measured for Big[+] is larger than those conferred by >30% of non-essential genes (*Breslow et al., 2008*), and the vast majority of natural genetic variants that have been quantified (*Jakobson and Jarosz, 2019*; *Jakobson et al., 2019*; *Sharon et al., 2018*; *She and Jarosz, 2018*).

Given the close link between proliferation and aging in many organisms (*Bitto et al., 2015*), we investigated whether Big[+] also influenced lifespan. Studies of aging in budding yeast have led to the discovery of numerous genes with conserved roles in aging of metazoans (*Kaeberlein, 2010*). These studies have measured two types of aging, both of which we tested here—chronological lifespan and replicative lifespan (RLS; *Longo et al., 2012*). Chronological lifespan is a measure of post-mitotic viability, during which cells cease division under starvation conditions until nutrients become available again. These conditions occur commonly in the natural ecology of budding yeast (*Landry et al., 2006*). To investigate, we aged cultures of naïve cells and genetically identical cells harboring Big[+] (*Figure 1—figure supplement 1B*). Over the course of 80 days, Big[+] cells had progressively lower viability than matched isogenic-naïve controls (*Figure 1C*).

RLS measures the number of cell divisions that yeast can undergo before death (*Figure 1—figure supplement 1B*). These experiments revealed a significant difference in replicative potential between Big[+] and naïve cells—the median survival rate is reduced by seven generations (*Figure 1D*, p<0.0001, Gehan–Breslow–Wilcoxon test, and *Figure 1—figure supplement 1C*). This degree of shortening is comparable to that of classic lifespan-altering mutants: yeast lacking *SIR3* or *SIR4* have reductions of ~3–4 generations, and yeast lacking *SIR2* by ~10–11 generations (*Kaeberlein et al., 2005*; *Kaeberlein et al., 1999*). Thus, Big[+] cells harboring this Pus4-induced element exhibit both a significantly decreased chronological lifespan and RLS: they *die young*.

## Modeling trade-offs between proliferation and lifespan

We next investigated the adaptive value of the Big[+] phenotype by quantitatively modeling its fitness consequences in fluctuating environments, where committing to a single strategy can impose limits on the long-term fitness of a population. For example, when nutrient-rich periods tend to greatly exceed starvation periods, the rapid growth of the Big[+] phenotype might be favored, despite its die-young phenotype (*Figure 2A*). However, this same decision could be maladaptive if growth conditions

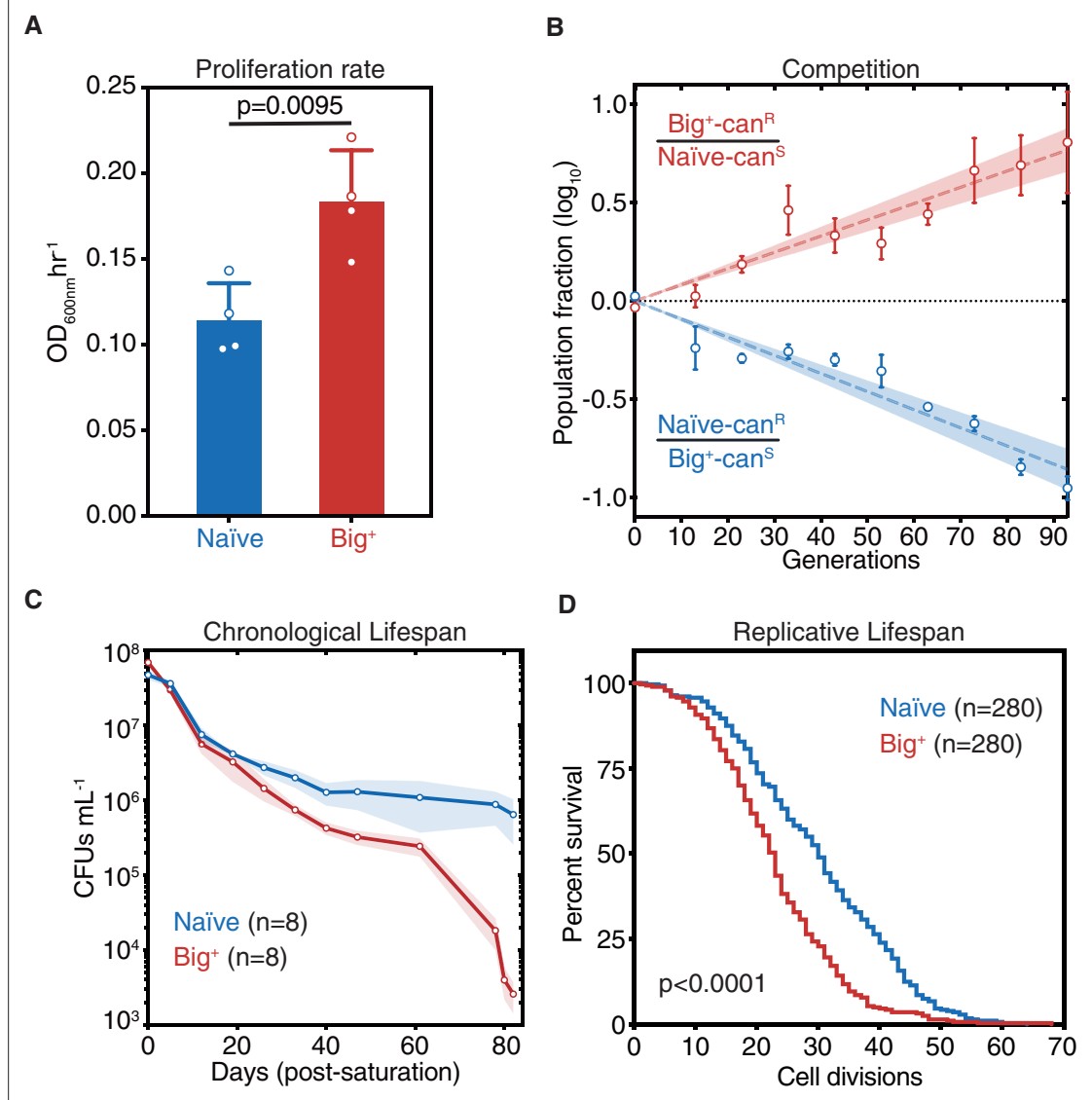

**Figure 1.** Cells bearing a prion-like epigenetic element live fast and die young. (**A**) Big+ cells proliferate faster than naïve cells. Bars represent the mean of four replicates of maximum growth rate in YPD medium (measured by the peak of the derivative of the growth data), error bars are standard deviation, p=0.0095, unpaired t-test. (**B**) In a direct growth competition, Big+ cells outcompete naïve cells. Raw data with standard error bars; trend line is dashed line showing shaded standard error; four replicates were performed for each competition. (**C**) Big+ cells have a reduced chronological lifespan (CLS). Cells were grown to saturation in rich medium and then transferred to nutrient-poor medium and allowed to age for up to 80 days. Periodically samples were replated onto rich medium to measure remaining viability via colony-forming units (CFUs). Thin lines are the average value from eight biological replicates with standard error represented by shading. (**D**) Big+ cells have a reduced replicative lifespan (RLS). Starting with virgin mother cells, at each cell division, daughter cells were separated, and the total number yielded was counted for each replicate. n = 280 per strain, combined from three independent experiments. p-value<0.0001, by Gehan–Breslow–Wilcoxon test. Median survival: naïve = 30 generations, Big+ = 23 generations.

The online version of this article includes the following figure supplement(s) for figure 1:

**Figure supplement 1.** Probing lifespan of Big+ cells.

skewed towards frequent and more extended periods of nutrient scarcity—where cells that grow slower and die older would instead have an advantage (*Figure 2B*).

To quantify these trade-offs, we considered a population in which individuals heritably adopted one of these two strategies (live-fast-die-young, *LF-DY*, like Big+ cells; or live-slow-die-old, *LS-DO*, like naïve cells), modeling their fates in alternating nutrient environments (*Figure 2C*). These simulations suggested that *LS-DO* cells should outcompete *LF-DY* cells when the periods of starvation are much longer than periods of nutrient abundance (*Figure 2—figure supplement 1A*). By contrast,

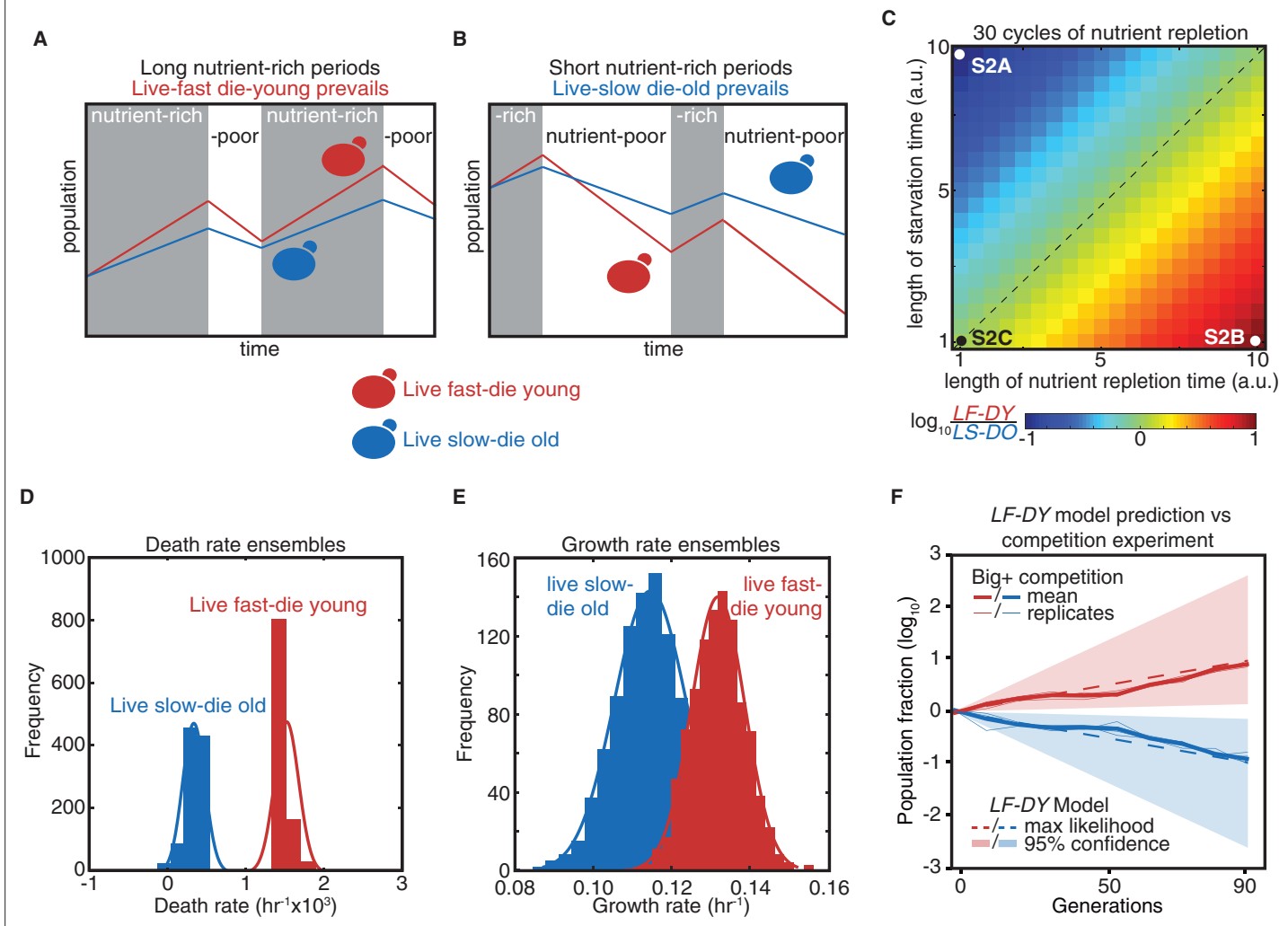

**Figure 2.** Modeling a reversible epigenetic live fast and die young strategy. (**A**) A live fast–die young epigenetic element is beneficial for survival in environments with regular, extended nutrient-rich periods. (**B**) A live slow–die old growth state is beneficial for survival during conditions of repeated and extended starvation. (**C**) Simulated final population fraction (ratio of live-fast-die-young [*LF-DY*] to live-slow-die-old [*LS-DO*]) after 30 cycles of nutrient repletion and starvation, assuming a 1% growth advantage, a 1% higher death rate, and equal starting population sizes. Note log scale. (**D**) Monte Carlo sampling of exponential decay constant and (**E**) exponential growth constant distributions used to generate the ensemble of simulations shown in (**F**). (**F**) Monte Carlo simulation (dashed lines) of growth competition between *LF-DY* to *LS-DO* cells under parameters sampled from experimental growth and lifespan measurements of the Big[+] element (*Figure 2D and E*). 95% confidence interval indicated by shaded areas. Shown in solid lines are the results of competitive growth between Big[+] and naïve cells as shown in *Figure 1B* (mean: bold line; n = 4 biological replicates: thin lines).

The online version of this article includes the following figure supplement(s) for figure 2:

**Figure supplement 1.** Modeling a reversible epigenetic live fast and die young strategy.

*LF-DY* cells come to dominate the simulated culture when periods of nutrient abundance are much longer than periods of starvation (*Figure 2—figure supplement 1B*). When periods of starvation and nutrient repletion are of equal duration, both populations are equally fit (*Figure 2—figure supplement 1C*). When we varied the growth advantage and lifespan cost of the *LF-DY* sub-population and the periods of nutrient availability and starvation, we noted that each phase space contained regimes in which *either* strategy could be advantageous (*Figure 2C* and *Figure 2—figure supplement 1D*).

Regimes in which either the *LF-DY* or *LS-DO* strategies would be strongly adaptive (and the other maladaptive) frequently arose under physiologically relevant environmental parameters. Oscillations within these regimes (i.e., between feast and famine) are common in nature (*Broach, 2012*), and withstanding them is essential for survival. Theory predicts that reversible epigenetic mechanisms, such

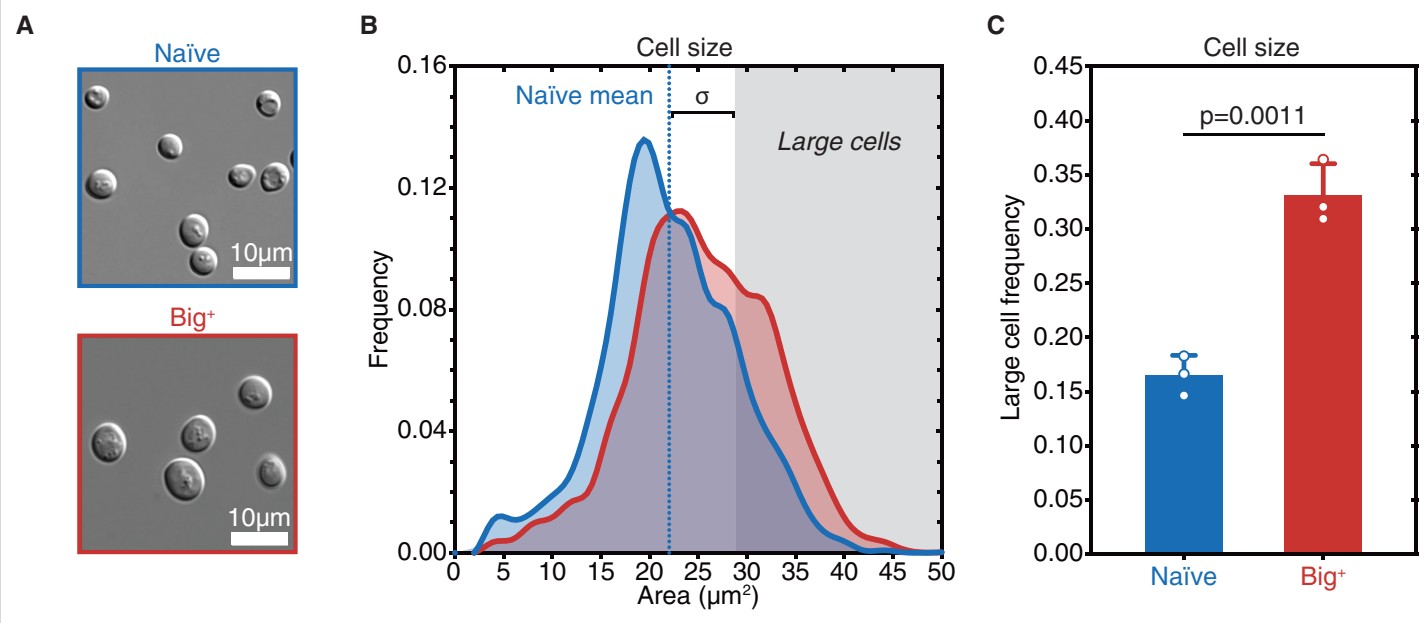

**Figure 3.** Big+ cells are large. (**A**) Micrographs of naïve and Big+ haploid yeast cells. (**B**) Cell size distributions for thousands of naïve and Big+ haploid cells (100% of distribution is shown, n = 4678 for naïve, n = 5501 for Big+, dotted line indicates naïve mean). Large cell threshold begins at one standard deviation above the naïve mean. (**C**) The frequency of haploid cells above the large cell threshold. Bars represent the mean of three replicate strains, for which thousands of cells are measured for each strain, error bars are standard deviation. p=0.0011, unpaired t-test.

The online version of this article includes the following figure supplement(s) for figure 3:

**Figure supplement 1.** The large cell phenotype of Big+ emerges during the growth of a culture.

as prions, could confer a selective advantage in fluctuating environments (**King and Masel, 2007**). Importantly, our model demonstrates that this advantage could derive not only from a trade-off between improved stress resistance and impaired growth in normal conditions but also from a growth advantage in times of plenty coupled with a disadvantage under stress.

Notably, the growth advantages we observed in our competition experiment were quantitatively consistent with Monte Carlo simulations sampled from our experimental measurements of the death rates (from chronological lifespan measurements, **Figure 2D**) and growth rates (from proliferation rate measurements, **Figure 2E**) of individual cultures in nutrient-starved and replete conditions, respectively (**Figure 2—figure supplement 1E**). That is, the adaptive advantages that we measured in competition were equivalent to those that we predicted for a hypothetical *LF-DY* population after dozens of generations (**Figure 2F**). Thus, selection on these properties could be alone sufficient to favor maintenance of the Big+ state under the conditions we examined.

## Big+ cells are large

Positive correlations between growth rate and cell size have long been noted (**Johnston et al., 1977**; **Schaechter et al., 1958**; **Su and O'Farrell, 1998**; **Turner et al., 2012**). To determine if the faster-growing Big+ cells were also larger, we examined them microscopically, employing a widely used image masking pipeline (**Carpenter et al., 2006**). This approach allowed us to measure the sizes of thousands of cells from multiple biological replicates and define size distributions for both naïve and Big+ populations.

During exponential growth, populations of Big+ cells had a similar size distribution to naïve control cells. However, as the cultures became denser, naïve control cells remained the same size whereas isogenic haploid Big+ cells became larger (**Figure 3A** and **Figure 3—figure supplement 1A**). To simplify these comparisons, we scored distributions by the fraction of very large cells that we observed (one standard deviation or larger than the mean naïve size; **Figure 3B**, n = 4678 and 5501 cells shown for naïve and Big+, respectively). This increase in mean area from 22.01 $\mu m^2$ to 25.36 $\mu m^2$ corresponds to a 23% larger volume (approximating the yeast cell as a sphere, naïve cell mean radius = 2.65 $\mu m$,

Big$^+$ cell mean radius = 2.84 μm). In Big$^+$ cultures 33.1% ± 2.8% of cells exceeded this threshold, whereas 16.5% ± 1.8% did in naïve cultures (p=0.0011 by unpaired t-test; *Figure 3C*). We note that this difference is not as large as that between haploid and diploid yeast; the naïve diploid mean area is 36.88 μm². Moreover, in mRNA-sequencing—discussed in more detail below—we observed expression profiles in Big$^+$ cells consistent with being haploid and mating type Mat a (like their naïve counterparts) (*Figure 3—figure supplement 1B and C*). We therefore conclude that the Big$^+$ state does not represent a simple autodiploidization.

## [*BIG$^+$*] is a prion transmitted through mating and meiosis

Big$^+$ was initially induced by transient *PUS4* overexpression in a screen to identify prion-like epigenetic elements (*Chakrabortee et al., 2016a*). We therefore tested whether the increased cell size associated with this state was transmitted through genetic crosses with the unusual patterns of inheritance that characterize prion-based phenotypes (*Brown and Lindquist, 2009*; *Cox, 1965*; *Wickner, 1994*). We began by mating the large haploid Big$^+$ cells to naïve haploids of the opposite mating type, selecting diploid cells, and measuring their size. The resulting diploids were significantly larger than those derived from control crosses with two naïve parents (*Figure 4A and B*), establishing that the trait is dominant.

We next investigated the meiotic inheritance of the large cell size trait. Because prions are not driven by changes in nucleic acid sequence, they have unusual patterns of inheritance that defy Mendel's laws. In addition to dominance in genetic crosses, prion-based traits can be passed to *all* progeny of meiosis, in contrast to DNA-based traits, which are inherited by half (*Figure 4—figure supplement 1A*; *Garcia and Jarosz, 2014*; *Li and Kowal, 2012*; *Liebman and Chernoff, 2012*; *Wickner, 2016*). We first mated control naïve cells to naïve cells of the opposite mating type, sporulated the resulting diploids, and dissected their meiotic progeny. We then grew clonal cultures of these haploid meiotic progeny and examined their size distributions, which were indistinguishable from their haploid-naïve parents (*Figure 4C*).

In contrast, all cultures derived from the meiotic progeny of Big$^+$× naïve crosses were large (*Figure 4C* and *Figure 4—figure supplement 1B*). This non-Mendelian pattern of inheritance differs strongly from the expected behavior for genetic mutants or chromatin-based epigenetic elements but is consistent with a prion-based mechanism of transmission. We therefore term this state '[*BIG$^+$*]' (with capital letters indicating dominance and brackets denoting its non-Mendelian pattern of segregation).

## [*BIG$^+$*] propagation requires the Hsp70 chaperone

In contrast to those driven by genetic mutations, the inheritance of prion-based phenotypes is strongly dependent upon the protein homeostasis network (*Garcia and Jarosz, 2014*; *Harvey et al., 2018*; *Liebman and Chernoff, 2012*; *Shorter and Lindquist, 2005*). As a consequence, *transient* inhibition of molecular chaperones can lead to the *permanent* elimination of prion-based traits. We therefore examined whether the large size of [*BIG$^+$*] cells also depended on protein chaperone activity (*Figure 4—figure supplement 1C*). Transient inhibition of the Hsp104 disaggregase, which regulates the inheritance of many amyloid prions (*Chernoff et al., 1995*; *Eaglestone et al., 2000*; *Halfmann et al., 2012*; *Shorter and Lindquist, 2004*), did not affect cell size in either naïve or [*BIG$^+$*] cells (*Figure 4D*). Transient inhibition of the Hsp90 foldase, which regulates the transmission of a different subset of prions (*Chakrabortee et al., 2016a*), also had no impact on [*BIG$^+$*] transmission (*Figure 4E*). By contrast, transient inhibition of Hsp70 via expression of a dominant-negative *SSA1*$^{K69M}$ allele (*Chakrabortee et al., 2016a*; *Jarosz et al., 2014*; *Lagaudriere-Gesbert et al., 2002*) caused [*BIG$^+$*] cells to lose their large size phenotype permanently (*Figure 4F*). Thus like other prions (*Brown and Lindquist, 2009*; *Chakrabortee et al., 2016a*; *Chakravarty et al., 2020*), and unlike genetic mutations, propagation of [*BIG$^+$*] is dependent on the activity of this ubiquitous molecular chaperone.

We note that Hsp70 expression drops dramatically as yeast reach saturation and begin to starve (*Werner-Washburne et al., 1989*; *Werner-Washburne and Craig, 1989*) and also decreases as cells age (*Janssens et al., 2015*). These are two scenarios in which [*BIG$^+$*] is disadvantageous. Therefore, environmental conditions that favor growth, during which Hsp70 is abundant, also favor prion propagation. By contrast, conditions known to reduce Hsp70 expression and thereby increase prion elimination are also those in which prion loss would be favored.

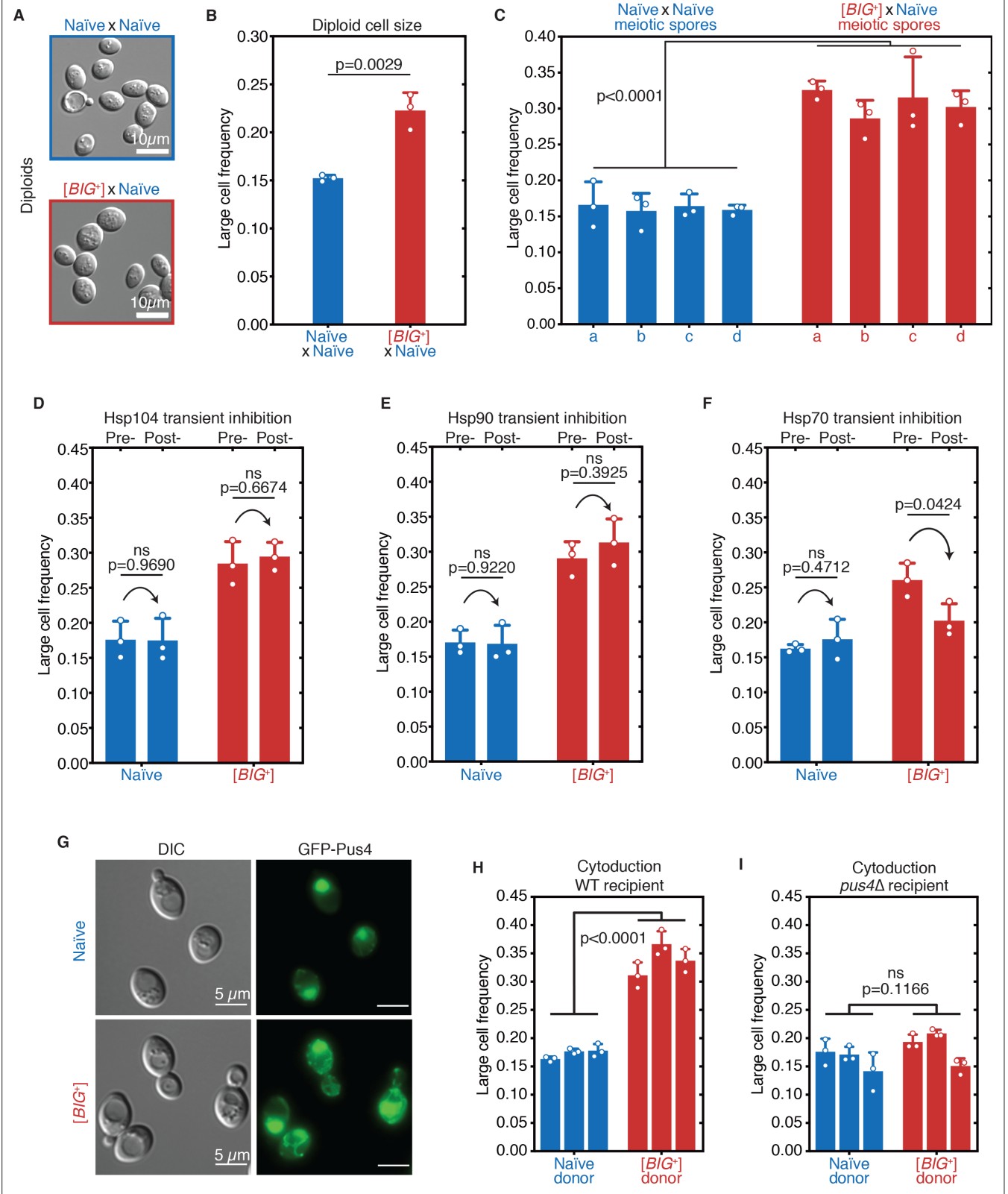

**Figure 4.** [*BIG*⁺] has prion-like patterns of inheritance. (**A**) Micrographs of diploid yeast cells resulting from crosses of naïve and naïve parents, or naïve and [*BIG*⁺] parents. (**B**) The frequency of diploid cells above the large cell threshold. Bars represent the mean of three replicate strains, for which thousands of cells were measured for each strain, error bars are standard deviation. p=0.0029, unpaired t-test. (**C**) Inheritance of large cell frequency to all meiotic spores. Bars represent the mean frequency of cells above the large cell threshold from three replicates, for which thousands of cells were

*Figure 4 continued on next page*

*Figure 4 continued*

measured for each replicate, error bars are standard deviation. Difference between the means of four tetrad spores between naïve and [*BIG*+], p<0.0001, unpaired t-test. (**D**) Transient inhibition of Hsp104 chaperone activity using guanidinium hydrochloride (GdnHCl) does not heritably alter the cell size trait. Bars represent the mean frequency of cells above the large cell threshold from three replicates, for which thousands of cells were measured for each replicate, error bars are standard deviation. Control samples (left bars of each pair) were propagated in parallel on nutrient-matched agar plates not containing GdnHCl. Post-inhibition represents strains subjected to GdnHCl treatment followed by recovery prior to cell size measurements (Materials and methods). Naïve p=0.9690, [*BIG*+] p=0.6674; unpaired t-test for both. (**E**) Transient inhibition of Hsp90 chaperone activity using radicicol does not heritably alter the cell size trait. Bars represent the mean frequency of cells above the large cell threshold from three replicates, for which thousands of cells were measured for each replicate, error bars are standard deviation. Control samples (left bars of each pair) were propagated in parallel on nutrient-matched agar plates not containing radicicol. Post-inhibition represents strains subjected to radicicol treatment followed by recovery prior to cell size measurements (Materials and methods). Naïve p=0.9220, [*BIG*+] p=0.3925; unpaired t-test for both. (**F**) Transient inhibition of Hsp70 chaperone activity by expression of a dominant-negative allele of *SSA1* permanently eliminates the [*BIG*+] cell size trait. Bars represent the mean frequency of cells above the large cell threshold from three replicates, for which thousands of cells were measured for each replicate, error bars are standard deviation. Control samples (left bars of each pair) did not contain the *SSA1*$^{K69M}$ constitutive expression plasmid but were propagated in parallel on non-dropout but otherwise nutrient-matched agar plates. Post-inhibition represents strains subjected to plasmid expression followed by plasmid removal and recovery prior to cell size measurements (Materials and methods). Naïve p=0.4712, [*BIG*+] p=0.0424; unpaired t-test for both. (**G**) The expression pattern of Pus4 is altered in [*BIG*+] cells. (**H**) [*BIG*+] can be transmitted via cytoduction into a wild-type recipient cell, consistent with a prion-based mechanism. Each bar represents the mean frequency of cells above the large cell threshold from three biological replicates, for which thousands of cells were measured for each replicate, error bars are standard deviation. Bars for three independent cytoductants are shown for each donor strain. Difference between the means of the three cytoductants between naïve and [*BIG*+] donors: p<0.0001, unpaired t-test. (**I**) [*BIG*+] is not transmitted via cytoduction into a *pus4Δ* recipient cell, indicating that prion transmission depends on continuous endogenous expression of Pus4. Each bar represents the mean frequency of cells above the large cell threshold from three biological replicates, for which thousands of cells were measured for each replicate, error bars are standard deviation. Bars for three independent cytoductants are shown for each donor strain. Difference between the means of the three cytoductants between naïve and [*BIG*+] donors: p=0.1166, unpaired t-test.

The online version of this article includes the following figure supplement(s) for figure 4:

**Figure supplement 1.** Tests for prion-like patterns of inheritance and dependence on Pus4 for [*BIG*+].

## Pus4 protein has a different expression pattern in [*BIG*+] cells

Prion states often impact the localization of the proteins that encode them. To visualize endogenous levels of Pus4 expression in naïve and [*BIG*+] cells, we employed a strain in which an N-terminal GFP tag was appended at the endogenous *PUS4* locus, and therefore subject to endogenous regulation (i.e., not overexpressed; *Weill et al., 2018*; *Yofe et al., 2016*). We did not observe large fluorescent foci typical of amyloid prions (*Alberti et al., 2009*), as might be expected given that Pus4 is not enriched in glutamine and asparagine residues typically abundant in amyloid prion proteins (*Chakrabortee et al., 2016a*). We did, however, observe altered localization of Pus4. In naïve cells, Pus4 consistently localized to the nucleolus, as has been previously reported (*Huh et al., 2003*). In [*BIG*+] cells, the Pus4 protein was also present in the nucleolus, but we also observed substantial fluorescence in a fragmented network throughout the cytoplasm (*Figure 4G*). Although a high-resolution structure awaits determination, our data are consistent with an altered state of Pus4 in [*BIG*+] cells.

## Endogenous Pus4 is required for propagation of [*BIG*+]

Although [*BIG*+] was induced by a transient increase in Pus4 abundance and was stable after eliminating the inducing plasmid, we wanted to exclude the possibility that this prior overexpression event might have established a positive feedback loop leading to an enduring increase in Pus4 levels. We therefore constructed naïve and [*BIG*+] strains with a seamless N-terminal 3X-FLAG tag endogenously encoded at the *PUS4* locus. Using immunoblots to detect the FLAG epitope in naïve and [*BIG*+] cells, we observed equivalent Pus4 levels, indicating that the phenotypes we observed in [*BIG*+] cells were not simply due to increased expression of this RNA-modifying enzyme (*Figure 4—figure supplement 1D*).

Prion proteins can be inherited through the cytoplasm and do not require the exchange of genetic material for propagation. To test this, we performed a cytoduction, in which we mated [*BIG*+] cells to naïve recipient cells of the opposite mating type carrying the *kar1-Δ15* mutation. Upon mating, this mutation prevents nuclear fusion, permitting the mixing of cytoplasm, but not nuclei, between donor and recipient cells (*Figure 4—figure supplement 1E*, Materials and methods; *Vallen et al., 1992*). The transfer of [*BIG*+] cytoplasm into naïve recipient cells resulted in the transfer of the [*BIG*+] cell size

phenotype (*Figure 4H*). However, this was only the case for wild-type recipients. Naïve recipients lacking *PUS4* did not acquire the [*BIG+*] cell size phenotype (*Figure 4I*).

It remained formally possible that a multi-protein prion state could be maintained by other cellular factors, even if prion-based phenotypes depended on Pus4. Therefore, we tested whether transient loss of *PUS4* was sufficient to eliminate [*BIG+*] permanently. We first deleted *PUS4* from [*BIG+*] and naïve cells. Upon *PUS4* deletion, the sizes of the mutants derived from naïve and [*BIG+*] parents were equivalent (*Figure 4—figure supplement 1F*). Together with our cytoduction data, this suggested that continuous production of Pus4 is required to maintain this trait. Furthermore, *PUS4* deletion did not increase the size of naïve cells, suggesting that [*BIG+*] does not inactivate Pus4, in contrast to many well-characterized prions that phenocopy loss-of-function alleles of their underlying proteins (*Byers and Jarosz, 2014*). Finally, we restored the *PUS4* gene to its native locus in these same cells by homology-directed integration. Even after the re-introduction of *PUS4*, the size distributions of both populations remained equivalent (*Figure 4—figure supplement 1F*). Thus, both the expression and the propagation of the [*BIG+*] phenotype require the continual presence of a *PUS4* gene product.

These various lines of evidence—transmission to all meiotic progeny, dependence on molecular chaperones, altered protein localization, and requirement for continuous expression of the protein that initiated the epigenetic trait—lead us to propose that [*BIG+*] is a protein-based element of inheritance, a prion, formed by the Pus4 pseudouridine synthase.

## Pus4-dependent pseudouridylation of tRNAs is maintained in [*BIG+*] cells

Because loss of *PUS4* did not phenocopy [*BIG+*] but did block propagation of the prion, we wondered whether the catalytic activity of Pus4 was maintained in [*BIG+*] cells. To measure pseudouridylation in naïve and [*BIG+*] cells, we employed a qPCR-based method that capitalizes on the enhanced susceptibility of pseudouridine to reacting with CMC (1-*c*yclohexyl-(2-*m*orpholinoethyl)*c*arbodiimide metho-p-toluene sulfonate); *Figure 5A*; *Lei and Yi, 2017*. When pseudouridines are 'labeled' by CMC, and these RNAs are used as templates for replication by reverse transcriptase, the enzyme generates nucleotide deletions and other mutations at CMC-labeled sites that can be detected by differences in the melting temperature of the derived nucleic acid duplexes (as compared to non-pseudouridylated or unlabeled controls). Using this approach, we examined the pseudouridylation of an archetypical Pus4 substrate, tRNA$_{AGC}$ (Ala). When a pseudouridine is present and CMC is added, the melting curve shifts relative to an unlabeled control (no CMC). We observed a similar leftward shift in melting curves for both naïve and [*BIG+*] cells (*Figure 5B–D*).

In contrast, control cells lacking the Pus4 protein, and therefore lacking pseudouridylation at U55, did not produce this shift (*Figure 5B–D*). Because tRNAs bear pseudouridines at other positions—catalyzed by different pseudouridine synthase enzymes—which could potentially interfere with our analysis of Pus4 activity, we also directly examined the U55 position. We cloned the PCR products into plasmids and then used Sanger sequencing to identify positions with mutations indicating a CMC adduct. Within the T-arm stem-loop of tRNA$_{AGC}$ (*Figure 5—figure supplement 1A*), we saw mutations originating at U55 in CMC-labeled samples from both naïve and [*BIG+*] cells, but not in unlabeled samples or *pus4Δ* samples (*Figure 5—figure supplement 1B*). These data establish that Pus4-dependent modification of tRNAs at U55 is maintained in [*BIG+*] cells.

## Relative RNA levels are nearly unchanged in [*BIG+*] cells

In addition to its ubiquitous pseudouridylation activity on all tRNAs, Pus4 also modifies some mRNAs (*Carlile et al., 2014*; *Lovejoy et al., 2014*; *Schwartz et al., 2014*), which may impact mRNA stability (*Zhao et al., 2017*). We performed mRNA sequencing to discern whether the phenotypes of [*BIG+*] cells might be due to relative changes in mRNA levels. We grew naïve and [*BIG+*] cultures in YPD medium until late exponential phase, extracted total RNA, and enriched these samples for polyadenylated RNAs. Comparing four replicates each of naïve and [*BIG+*], the expression levels of only 47 genes changed significantly (28 decreased in expression and 19 increased as estimated using Kallisto and Sleuth; *Bray et al., 2016*; *Pimentel et al., 2017*; Benjamini–Hochberg-corrected $q$-value < 0.05 from the likelihood ratio test and |log$_2$ fold-change| > 1) (*Supplementary file 2*). Spike-in controls demonstrated quantitative scaling in our estimates of gene expression across a broad range of transcript abundance (*Figure 5—figure supplement 1C*). The upregulated genes in [*BIG+*] exhibited only

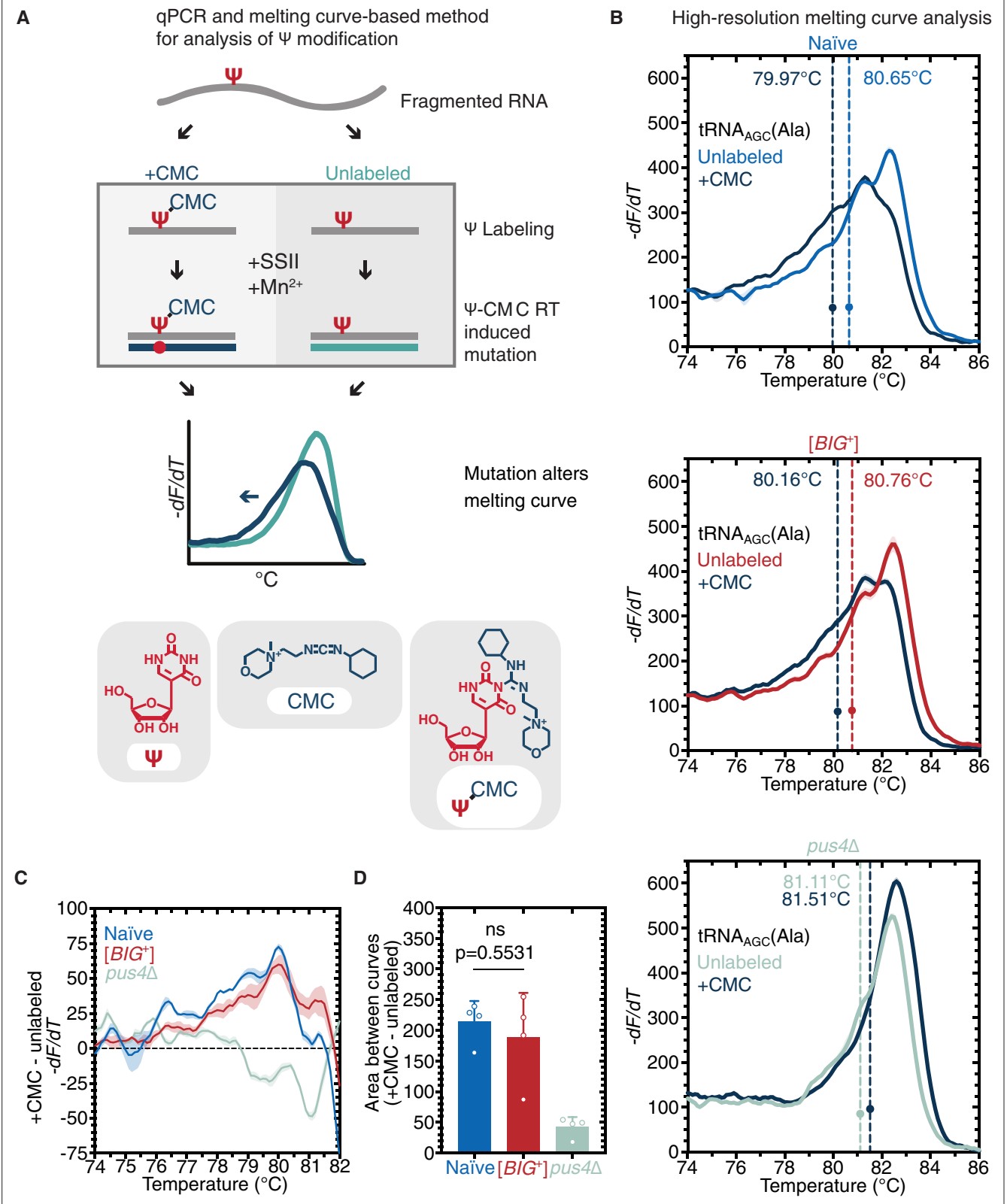

**Figure 5.** Pus4 activity is maintained in [*BIG*⁺]. (**A**) Radiolabeling-free, qPCR-based method for locus-specific pseudouridine detection. Illustration is adapted from Scheme 1 of **Lei and Yi, 2017**. (**B**) High-resolution melting curve analysis demonstrates that Pus4-dependent pseudouridylation of tRNA_AGC (Ala) is maintained in [*BIG*⁺] cells but not in cells that do not contain Pus4p. Top panel: naïve samples CMC-labeled (dark blue) or -unlabeled (light blue). Middle panel: [*BIG*⁺] samples CMC-labeled (dark blue) or -unlabeled (red). Bottom panel: *pus4Δ* samples CMC-labeled (dark blue) or

*Figure 5 continued on next page*

*Figure 5 continued*

-unlabeled (turquoise). Solid lines representing melting curves are the mean of four replicates, with shaded areas representing the standard error of the mean. Dots mark the geometric center of four replicates, bisected by a dashed line with a shaded area representing the standard error of the mean. The melting temperature (Tm) of this point is also displayed. The leftward shift of +CMC curves in naïve and [*BIG*+] but not *pus4Δ* samples indicates Pus4-dependent pseudouridylation of U55. (**C**) The difference in melting temperature behavior (*dF/dT*) between CMC-labeled and CMC-unlabeled tRNA_{AGC} (Ala) amplicons is similar between [*BIG*+] cells and naïve cells, suggesting that Pus4-dependent $\Psi$ is present at position 55 in tRNA_{AGC} (Ala) in [*BIG*+] cells. Solid line represents the mean of four replicates, with shaded areas showing standard error of the mean. (**D**) The difference in area between the melting curves of CMC-labeled and CMC-unlabeled tRNA_{AGC} (Ala) amplicons is similar between [*BIG*+] cells and naïve cells, suggesting that Pus4-dependent $\Psi$ is present in [*BIG*+] cells. Bars represent the mean of four replicates, error bars indicate standard deviation, p=0.5531, unpaired t-test.

The online version of this article includes the following figure supplement(s) for figure 5:

**Figure supplement 1.** tRNA pseudouridylation and quantitative scaling of gene expression in mRNA sequencing.

a modest enrichment for water transport (two aquaporin genes), while the downregulated genes were enriched for cell wall and plasma membrane components (*Supplementary file 2*; *Szklarczyk et al., 2019*), perhaps because larger [*BIG*+] cells have less surface area per unit volume. We conclude that the major effects of [*BIG*+] during exponential growth (e.g., on growth rate and RLS) likely do not occur via changes to steady-state mRNA levels.

To investigate whether the relative abundance of tRNAs is perturbed in [*BIG*+] cells, we ran total RNA from naïve and [*BIG*+] cells (grown to late-exponential phase) on a nucleic acid fragment analyzer and quantified tRNA levels relative to a similarly abundant RNA that is not a target of Pus4, 5.8 S rRNA (158 nt) (*Figure 5—figure supplement 1D*). We observed no significant differences. While we cannot exclude other effects on tRNA function or the relative abundance of particular tRNAs, our data show that bulk tRNA abundance differences are likely also not responsible for the phenotypes of actively growing cells containing [*BIG*+].

## [*BIG*+] cells are resistant to inhibition of protein synthesis

Because the increased cell size and proliferation and reduced lifespan of [*BIG*+] cells occur in the absence of major alterations to relative abundances of mRNA or tRNA, we wondered whether a change in protein synthesis might be responsible. To test this, we employed two inhibitors: (1) cycloheximide, an inhibitor of translational elongation (*Baliga et al., 1969*; *McKeehan and Hardesty, 1969*), and (2) rapamycin, a natural product macrolide that inhibits the TOR kinase, blocking a conserved signaling cascade that promotes protein synthesis (*Beretta et al., 1996*; *Chung et al., 1992*; *Kuo et al., 1992*; *Price et al., 1992*; *Urban et al., 2007*).

[*BIG*+] cells grew nearly twofold better than naïve cells in a sub-inhibitory concentration of cyclo-heximide that was sufficient to impair proliferation (0.0 1 µg/mL, in which naïve cells needed 23.6 hours to reach their maximum proliferaton rate vs. 21.3 hours in YPD; for comparison, 100 µg/mL is typically used for experiments that rapidly and completely arrest translation; *Figure 6A*). [*BIG*+] cells also proliferated faster in a concentration of rapamycin (10 µM) that inhibited growth (*Figure 6B*), suggesting that the pathway may be more active in cells harboring the prion. We found additional evidence in support of this hypothesis by examining the localization of Par32 (*p*hosphorylated *a*fter *r*apamycin 32); the higher ratio of membrane-localized Par32 that we observed in [*BIG*+] cells is consis-tent with a more active TOR pathway (*Figure 6—figure supplement 1A*; *Varlakhanova et al., 2018*). These latter observations also provide a potential explanation for the decreased longevity of [*BIG*+] cells: loss of TOR pathway function is associated with extended lifespan in yeast (*Dikicioglu et al., 2018*; *Fabrizio et al., 2001*; *Powers et al., 2006*) and many other organisms including nematodes, fruit flies, and mice (*Bjedov et al., 2010*; *Harrison et al., 2009*; *Robida-Stubbs et al., 2012*).

We next investigated whether these inhibitors impacted the size of [*BIG*+] cells. When grown in cycloheximide or rapamycin to saturation, [*BIG*+] cells were no longer large compared to naïve controls (*Figure 6C and D*). Thus, unperturbed translation is necessary for the increased size of [*BIG*+] cells. These data could be explained by the inhibitors masking expression of the large cell trait or by reversion of the [*BIG*+] prion. To distinguish between these possibilities, we sub-cultured cells that had been treated with cycloheximide or rapamycin and allowed them to recover in rich medium for several dozen generations. We then examined their size distributions. Populations derived from [*BIG*+] ancestors were once again significantly larger than naïve controls that we subjected to the same

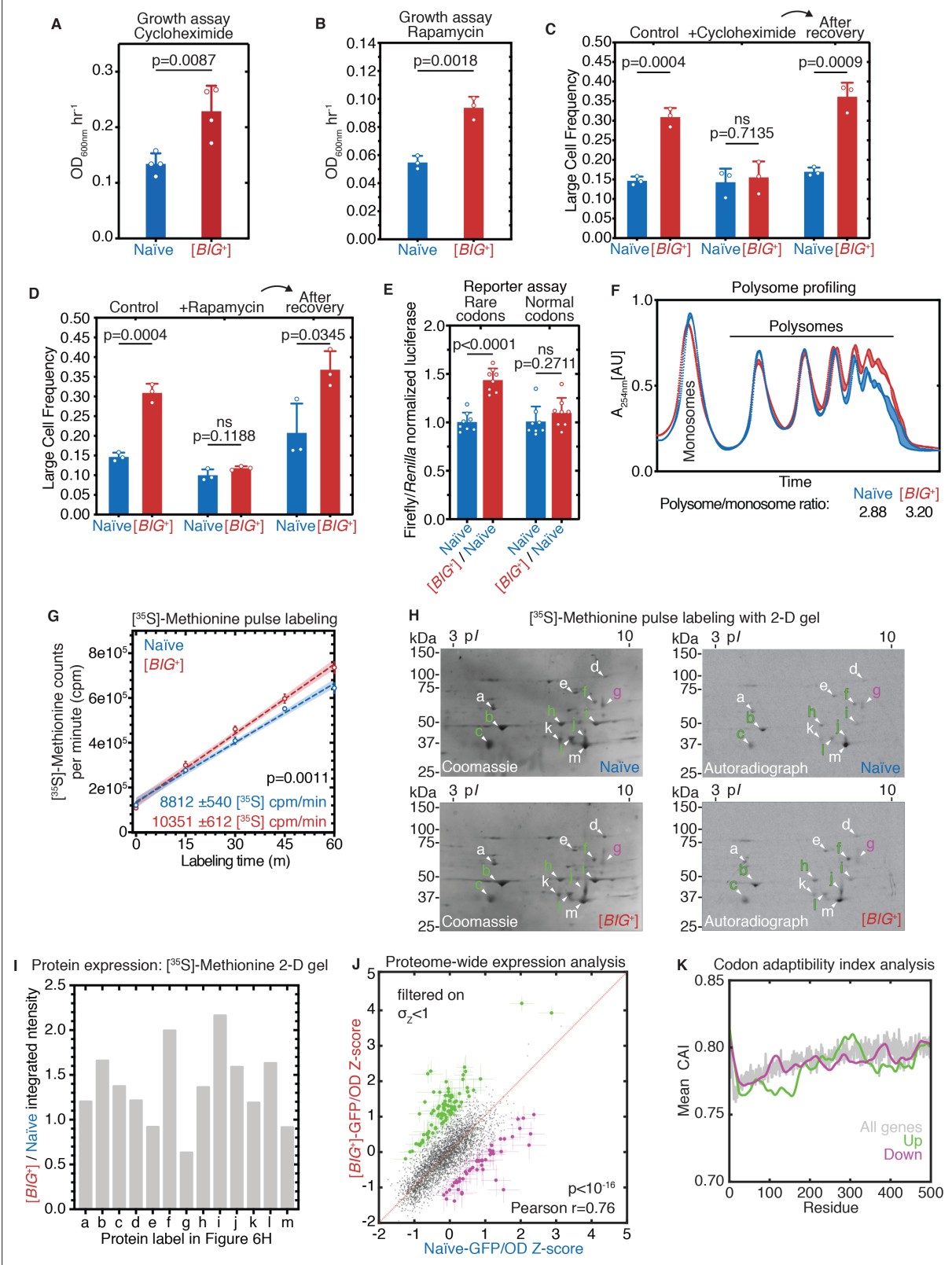

**Figure 6.** [*BIG+*] has altered protein synthesis. (**A**) [*BIG+*] cells are resistant to translation elongation inhibitor cycloheximide. Bars represent the mean of the maximum growth rate in YPD + cycloheximide (measured by the peak of the derivative of the growth data) of four replicates, error bars are standard deviation, p=0.0087, unpaired t-test. (**B**) [*BIG+*] cells are resistant to TOR inhibitor rapamycin. Bars represent the mean of the maximum growth rate in YPD + rapamycin (measured by the peak of the derivative of the growth data) of three replicates, error bars are standard deviation, p=0.0018,

*Figure 6 continued on next page*

*Figure 6 continued*

unpaired t-test. (**C**) [*BIG*⁺] cells grown in cycloheximide (0.05 µg/mL) are not larger than naïve cells. However, after recovery, they regain this phenotype. After treatment, cells were sub-cultured in YPD for ~75 generations before re-measuring the size in the absence of stress (see Materials and methods). Bars represent the mean frequency of cells above the large cell threshold from three replicates, for which thousands of cells were measured for each replicate, error bars are standard deviation. Difference between the means of naïve and [*BIG*⁺]: YPD control p=0.0004; YPD + cycloheximide p=0.7135; YPD after recovery p=0.0009; unpaired t-test for all. (**D**) [*BIG*⁺] cells grown in rapamycin are not larger than naïve cells. However, after recovery they regain this phenotype. After treatment, cells were subcultured in YPD for ~75 generations before re-measuring the size in the absence of stress (see Materials and methods). Bars represent the mean frequency of cells above the large cell threshold from three replicates, for which thousands of cells were measured for each replicate, error bars are standard deviation. Difference between the means of naïve and [*BIG*⁺]: YPD control p=0.0004 (same data presented in *Figure 6C* as experiments were done in parallel); YPD + rapamycin p=0.1188; YPD after recovery p=0.0345; unpaired t-test for all. (**E**) [*BIG*⁺] meiotic progeny translate more of a firefly luciferase reporter containing rare codons than naïve meiotic progeny do. This effect is not seen in an mRNA variant that encodes an identical protein but contains codons more frequently used in yeast. For each variant, normalized luciferase values in naïve cells are set to 1.0 to make presentation straightforward; however, the normal codon variant of firefly produces approximately five times more luciferase activity than the rare codon variant (data not shown; *Chu et al., 2014*). Bars represent mean normalized luciferase values (an invariable *Renilla* luciferase gene is co-expressed from the same plasmid) from eight replicates: rare codons p<0.0001, normal codons p=0.2711, unpaired t-tests for both. (**F**) [*BIG*⁺] cells have more polysomes than naïve cells, as measured by polysome profiling. Lines for two technical replicates for each sample are shown, with the area between them shaded. Ratios (average of two technical replicates) were calculated by taking the lowest point between the monosome and disome peak as zero, and then calculating the ratios of the areas under the sum of the polysome peaks to that under the monosome peaks. (**G**) Using [³⁵S]-methionine pulse labeling, [*BIG*⁺] cells have an increased rate of translation relative to naïve cells (slope line fit: p=0.0011). Lines for three biological replicates for each sample are shown, with the area between them shaded. Timepoints were taken at 1, 15, 30, 45, and 60 min. (**H**) Separation of total protein by 2-D gel electrophoresis from cells pulse-labeled for 1 hr with [³⁵S]-methionine demonstrates that many proteins are synthesized at a faster rate in [*BIG*⁺] cells, few at a slower rate, and some at an indistinguishably different rate, constituting an altered program of protein synthesis. For each gel, a Coomassie stain showing total protein and an autoradiography exposure newly synthesized protein are shown with a sampling of 13 distinguishable proteins labeled. Letters colored green or violet mark proteins whose expression was increased or decreased >1.25 -fold, respectively, in the analysis presented in the next panel. (**I**) Relative protein levels measured by dot intensities from autoradiographs shown in *Figure 6H*. Shown are the values for 13 proteins quantified in both naïve and [*BIG*⁺] [³⁵S]-methionine pulse-labeled samples. (**J**) Plot showing proteome-wide GFP::protein fusion expression in [*BIG*⁺] cells compared to naïve cells, highlighting ~130 proteins whose levels change. Each dot represents the mean of quadruplicate measurements of a single protein in naïve or [*BIG*⁺] cells: black dots are proteins that did not change significantly as measured by Z-score change of less than 1.0 ($\sigma_Z < 1$); green dots are protein fusions with higher fluorescence in [*BIG*⁺] cells; violet dots are protein fusions with lower fluorescence in [*BIG*⁺] cells. For colored dots, the standard error of the mean is shown for both measurements from four biological replicates each. Pearson correlation of naïve and [*BIG*⁺] cells, $r = 0.76$, and p<10⁻¹⁶ indicates that most proteins have correlated expression levels. $OD_{600}$ was adjusted based on known blank wells, and the GFP/$OD_{600}$ measurements were normalized by Z-score ($[x_i-\mu]/\sigma$) within the naïve and [*BIG*⁺] populations independently. (**K**) Plot showing the protein residue number vs. the mean codon adaptation index (CAI) for all measured GFP-tagged proteins (gray line) and proteins whose levels were increased (green line) or decreased (violet line) in [*BIG*⁺] relative to naïve cells in the proteome-wide screen. Proteins whose levels were elevated in [*BIG*⁺] relative to naïve cells have a lower mean CAI in the 5' ends of their mRNAs.

The online version of this article includes the following figure supplement(s) for figure 6:

**Figure supplement 1.** Altered cell cycle and translation in [*BIG*⁺].

propagation regime (*Figure 6C and D*). We conclude that [*BIG*⁺] depends on protein synthesis to augment cell size but that the prion is stable to transient perturbations in translation.

## [*BIG*⁺] cells have altered protein synthesis

Translation is rate-limiting for growth in nutrient-rich conditions (*Kafri et al., 2016*). We observed enhanced growth of [*BIG*⁺] cells in rich YPD medium (*Figure 1A and B*). However, in synthetic defined medium with identical carbon source abundance (2% glucose) but fewer amino acids, nucleosides, and other nutrients for optimum growth (SD-CSM), [*BIG*⁺] cells did not grow faster than naïve controls (*Figure 6—figure supplement 1B*). These data, combined with the resistance of [*PRION*⁺] cells to cycloheximide and rapamycin, suggested that protein synthesis might be enhanced in [*BIG*⁺].

Because a major component of translational regulation is the efficiency with which the ribosome translates each mRNA, we examined the impact of [*BIG*⁺] on mRNAs encoded with different codons using luciferase reporter assays. We transformed [*BIG*⁺] cells and isogenic naïve control cells with dual-luciferase plasmids encoding both *Renilla* and firefly luciferase genes. We tested two versions of the firefly luciferase gene: the first contained the standard suite of firefly mRNA codons; the second produced an identical protein product, but via codons that are rarer in *S. cerevisiae*, reducing steady-state protein levels by approximately five fold (*Chu et al., 2014*). Each plasmid contained an identical *Renilla* luciferase gene with its natural set of codons that served as an internal control (normalizing for copy number and general transcriptional activity). The firefly reporter with normal codons did not

produce more luciferase activity in [*BIG*+] cells than in naïve cells (*Figure 6E*). However, the firefly reporter with rarer codons produced normalized luciferase levels approximately 50% higher in [*BIG*+] cells than in isogenic naïve control cells (*Figure 6E*). These data suggest that [*BIG*+] cells may enhance the translation of some mRNAs, especially those containing a greater frequency of rare codons. We observed these effects in meiotic spores from naïve × [*BIG*+] crosses (*Figure 6E*) as well as in the original [*BIG*+] isolates (*Figure 6—figure supplement 1C*), showing that the altered translation phenotype, like cell size, is inherited by all progeny of meiosis.

We next investigated whether [*BIG*+] had global effects on the proteome by isolating total protein from cells harboring the prion and naïve controls. We reproducibly obtained more total protein per cell from the [*BIG*+] cultures (*Figure 6—figure supplement 1D*). We next loaded an *equal* mass of protein lysate from naïve and [*BIG*+] cells onto a denaturing polyacrylamide gel and separated them by electrophoresis. Coomassie staining of these gels showed no pronounced differences in banding patterns (*Figure 6—figure supplement 1E*), suggesting that [*BIG*+] does not exert substantial changes on the relative levels of the most highly expressed proteins, which are known to be efficiently translated (*Gingold and Pilpel, 2011*; *Plotkin and Kudla, 2011*).

We next performed polysome gradient analysis to assess ribosome density on mRNAs, a measure of global translation activity. We observed no change in monosomes or disomes in [*BIG*+] samples compared to naïve controls. However, polysomes—which are responsible for most protein synthesis (*Noll, 2008*; *Warner and Knopf, 2002*)—were increased in [*BIG*+] cells relative to naïve controls, with the polysome-to-monosome ratio increasing from ~2.9 to 3.2 (*Figure 6F*).

To further investigate translation in [*BIG*+] cells, we performed a pulse-labeling experiment to measure the rate of newly synthesized protein. We grew naïve and [*BIG*+] cells in medium containing [$^{35}$S]-methionine and harvested protein lysates every 15 min for 1 hr, and then measured the extent of radiolabel incorporation by scintillation counting. These experiments established that the rate of newly synthesized total protein was higher in [*BIG*+] cells (10,179 cpm/min) compared to naïve cells (8812 cpm/min) (*Figure 6G*; p<0.0011, unpaired t-test). In summary, we found that [*BIG*+] cells have a higher translation rate, which can increase total protein output, including the levels of some proteins translated from mRNAs enriched with codons generally associated with lower translation efficiency.

## [*BIG*+] reduces time spent in the G1 phase of the cell cycle

Conditions that enhance protein synthesis also tend to reduce the fraction of time that cells spend in the G1 stage of the cell cycle (*Jorgensen and Tyers, 2004*). This is because commitment to S phase entry—budding of a daughter yeast cell—depends on sufficient production of proteins needed to replicate the genome and essential cellular structures. Accelerating the production of these factors can thus shorten this period. We measured the fraction of naïve and [*BIG*+] cells in the G1 phase of the cell cycle by counting the fraction of unbudded cells (i.e., cells in G1). For naïve cells, 36.2% were in G1, whereas only 27.2% of [*BIG*+] cells were in G1, a ~25% reduction (*Figure 6—figure supplement 1F*, p=0.0017, unpaired t-test). These data suggest that the cell cycle checkpoint for progression to S phase remains intact, and that a shortened G1 stage in [*BIG*+] cells is consistent with their increased protein synthesis. [*BIG*+] cells are not larger during exponential phase growth (*Figure 3—figure supplement 1A*), suggesting that their cell size checkpoint remains intact. By contrast, cells in stationary phase, which do not have the nutrient content needed to progress to S phase, may continue to accumulate mass at a faster rate, contributing to their larger size.

## [*BIG*+] cells enact an altered translational program

Because relative mRNA expression levels were only subtly altered in [*BIG*+] cells during exponential growth phase, but multiple measures of translation were altered, we considered the possibility that the phenotypes of the prion might be due to changes in protein levels of particular open reading frames. We tested this idea using two methods: (1) 2-D gel electrophoresis on pulse-labeled, newly synthesized proteins, and (2) a screen for differential expression of over 5000 GFP-tagged proteins.

We pulse-labeled [$^{35}$S]-methionine into newly synthesized proteins for 1 hr and then separated total protein using 2-D gel electrophoresis. Although labeling for less time than a cell division limited the number of newly synthesized proteins we could measure, we chose this time to minimize the contributions from factors other than protein synthesis, such as degradation. We observed [$^{35}$S]-methionine counts of 9.7 cpm/μg and 11.5 cpm/μg protein from naïve and [*BIG*+] cells, respectively, indicating that

cells harboring the prion synthesized proteins faster than naïve cells (*Figure 6G*). We also analyzed the [$^{35}$S]-methionine signal from 13 proteins that we could separate by 2-D gel electrophoresis. In [*BIG*$^+$] cells, seven were increased, one was decreased, and five remained similar (*Figure 6H and I*). These data suggest that [*BIG*$^+$] cells have an altered translation program, with variations in even the most rapidly synthesized proteins.

To identify specific proteins with altered levels in prion-containing cells, we capitalized on the dominance of [*BIG*$^+$] in genetic crosses (*Figure 4B*), mating haploid cells harboring the prion to a genome-wide collection of superfolder-GFP gene fusions ('SWAT' library; ~ 5500 ORFs; fused seamlessly to retain endogenous promoter/regulatory elements; *Weill et al., 2018*). Equivalent matings between naïve strains and this genome-wide collection served as controls. To control for the potentially larger size of the [*BIG*$^+$] cells, we assessed protein levels in these diploid strains in terms of the relative GFP levels (normalized by OD$_{600}$ and Z-scored) within the naïve and [*BIG*$^+$] GFP-fusion collections separately. Mating and fluorescence measurements were performed in biological duplicate: the SWAT library was mated to two independent [*BIG*$^+$] isolates alongside naïve controls. The reported OD$_{600}$-normalized GFP levels are the mean of two technical duplicates of each biological duplicate. Protein levels measured in this way were generally well correlated between naïve and [*BIG*$^+$] strains (Pearson's $r$ = 0.76; p<10$^{-16}$; *Figure 6J*), in concordance with our results from electrophoresis of total cellular protein. However, many proteins were up- or downregulated in [*BIG*$^+$] cells.

Of the 4233 fusions whose abundance could be robustly quantified across the four replicates (Z-score < 1 for both naïve and [*BIG*$^+$]), ~130 were differentially expressed in [*BIG*$^+$] cells. Consistent with a bias toward enhanced translation, 81 were upregulated and 46 were downregulated (*Figure 6J* and *Supplementary file 3*). These proteins did not show any strong enrichment in physicochemical properties, nor were they enriched in Pus4-dependent pseudouridylation sites (see Appendix 1 for further discussion). We also did not observe a widespread increase in the expression of ribosomal proteins. We did, however, observe that proteins that were increased in [*BIG*$^+$] cells had a modest decrease in their codon adaptation index (CAI) at the 5' end of their mRNAs relative to all genes (*Figure 6K*), indicating that cells harboring the prion might more efficiently translate these messages. The dip in CAI that is typical in the 5' end of all genes, known as 'translational ramping,' is thought to reflect the bias toward translational control near the beginning of ORFs (*Frumkin et al., 2018*; *Tuller et al., 2010*). (Lower CAI can both reduce the speed of elongation by requiring rarer tRNAs and lead to differences in mRNA structure that might also affect initiation or elongation efficiency.) These data are consistent with our luciferase reporter data in which rarer codons throughout the message output more protein in [*BIG*$^+$] than in naïve cells (*Figure 6E* and *Figure 6—figure supplement 1C*), as well as the resistance of prion cells to the elongation inhibitor cycloheximide (*Figure 6A*).

Many hits were logically connected to the enhanced translation, increased size, and shortened lifespan of [*BIG*$^+$] cells. The 81 upregulated proteins included multiple ORFs whose deletions are associated with decreased cell size (such as *PHO5, MRPS28, KAP122,* and *SWE1*; *Harvey and Kellogg, 2003*; *Jorgensen et al., 2002*), increased chronological lifespan (*DIG2* and *UBX6*; *Garay et al., 2014*), and extended RLS (*ENO1, MUB1, PHO87, PGM2,* and *YRO2*; *McCormick et al., 2015*). Conversely, the 46 downregulated proteins included ORFs whose deletions are associated with increased cell size (*RNR4* and *RPL15B*; *Jorgensen et al., 2002*; *Perlstein et al., 2005*) and decreased chronological lifespan (including *BUD23, SMI1, MHP1,* and *CLG1*; *Garay et al., 2014*; *Marek and Korona, 2013*). Several proteins directly involved in translation control and ribosome biogenesis were also increased in [*BIG*$^+$] cells, including *FHL1, MRPS28, MRPS16, RPL24B, RPS7A,* and *UTP10* (*Figure 6—figure supplement 1G*). The [*BIG*$^+$]-regulated proteins also included 29 ORFs associated with differential sensitivity to rapamycin (including *SAS4, SNF1,* and *HCA4*; *Butcher et al., 2006*; *Dudley et al., 2005*; *Kapitzky et al., 2010*) and 14 ORFs that are known genetic or physical interactors with TOR1 (including *GCD14* and *PAR32*; *Krogan et al., 2006*; *Varlakhanova et al., 2018*). The functional breadth of these effects on the proteome and their logical connection to factors involved in the control of proliferation, cell size, and lifespan suggest that the phenotypes of [*BIG*$^+$] are likely derived from an altered translational regulation program that favors increased cell size and proliferation at the expense of lifespan.

## Pus4-dependent pseudouridylation of mRNAs in [*BIG*$^+$] cells

Our finding that pseudouridylation on tRNA was maintained at similar levels in [*BIG*$^+$] cells (*Figure 5B–D*) led us to wonder whether mRNA substrates of Pus4 were similarly modified. The

best-documented mRNA target of Pus4 is the translation elongation factor *TEF1/TEF2*—whose position U239 is robustly pseudouridylated in a Pus4-dependent manner (*Carlile et al., 2014*; *Lovejoy et al., 2014*; *Schwartz et al., 2014*). These paralogous genes encode identical copies of the eEF-1alpha translation elongation factor, which binds to aminoacylated tRNAs—specifically the T arm stem-loop that is pseudouridylated by Pus4 (*Dreher et al., 1999*)—and delivers them to the A-site of the ribosome during translation elongation (*Schirmaier and Philippsen, 1984*). We used three different methods to measure pseudouridylation of U239 in *TEF1/TEF2* from naïve, [*BIG+*], and *pus4Δ* cells. Sanger sequencing verified that the modified position was identical to that previously annotated in the literature (*Figure 7—figure supplement 1A*). Next, we used a quantitative RT-PCR-based method, known as CMC-RT and *l*igation-*a*ssisted *P*CR analysis of Ψ modification (CLAP) (*Figure 7A*; *Zhang et al., 2019*). We observed *TEF1/TEF2* U239 pseudouridylation in both naïve and [*BIG+*] cells, in contrast to very low/background levels in *pus4Δ* cells (*Figure 7B*). Finally, using the aforementioned qPCR-based method for detecting pseudouridylation (*Figure 5A*), we found additional evidence that *TEF1/TEF2* was pseudouridylated in our wild-type cells (both naïve and [*BIG+*]) in a Pus4-dependent manner (*Figure 7C*).

Most studies of yeast prions have characterized them as decreasing or eliminating activity (*Garcia and Jarosz, 2014*). However, we recently discovered one notable exception in which the [*SMAUG+*] prion can increase the activity of the protein that encodes it, Vts1 (*Chakravarty et al., 2020*). We thus examined if there was an altered level of pseudouridylation of *TEF1/TEF2* mRNA. If altered levels affected protein activity, this could be one possible mechanism linked to the altered translation program we found in [*BIG+*] cells. By quantifying differences in the melt curve shift after CMC labeling—an analysis made simpler for *TEF1/TEF2* than for tRNAs by the absence of other pseudouridylated positions flanking U239—we observed an increase in the signal of pseudouridylation in [*BIG+*] cells relative to naïve (*Figure 7D and E* and *Figure 7—figure supplement 1B*).

Together these data demonstrate that the catalytic function of Pus4 is retained in [*BIG+*] cells and may be enhanced for specific substrates, contrasting with the classical view of prions as being loss-of-function protein conformations. They also provide a novel example of how RNA modification can be epigenetically controlled.

## Epigenetic control of [*BIG+*]-like phenotypes in wild yeast

Finally, we tested whether protein-based epigenetic control of cell size, protein synthesis, or localization is present in wild yeast populations. Protein chaperones are essential regulators of prion propagation in wild strains, just as they are in laboratory strains (*Halfmann et al., 2012*; *Jarosz et al., 2014*). To block the passage of prions in wild yeasts, we transiently expressed a dominant-negative variant ($SSA1^{K69M}$) of Hsp70—the chaperone that is essential for [*BIG+*] propagation (*Figure 4F*)—in 20 wild *S. cerevisiae* strains isolated from various environments around the world (*Cubillos et al., 2009*; *Itakura et al., 2020*). We measured the size of cells before and after this chaperone curing and found that 20% (four isolates out of twenty)—'273614 N' (clinical, Newcastle, UK); 'L-1528' (fermentation, Cauquenes, Chile); 'DBVPG1373' (soil, Netherlands); 'BC187' (barrel fermentation, Napa Valley, California)—became smaller after curing (*Figure 8A*).

Given the wide array of genetic, epigenetic, and ecological diversity harbored in this collection, we also tested for three additional features associated with [*BIG+*] in laboratory strains: (1) increased protein synthesis, (2) differences in Pus4 localization between cured and uncured variants, and (3) a Pus4-dependent change in cell size.

First, to test whether the four isolates that became smaller upon curing had altered protein synthesis, we transformed them with the same luciferase reporters we tested in laboratory [*BIG+*] cells. We normalized the firefly reporters with variable codons to the *Renilla* control for cured strains and then normalized these to the corresponding uncured strain values. Two of the four isolates showed significant changes to translation (*Figure 8B*). BC187 yielded the largest change upon curing, an ~40% reduction in the translation of a firefly luciferase containing a 'normal' suite of codons, similar in magnitude to what we had observed with [*BIG+*] cells generated in the laboratory. Thus, since the loss of a chaperone-dependent epigenetic element reduced translational output from the reporter, the original, wild isolate (uncured) has higher protein synthesis, as we saw for [*BIG+*] in the BY4741 laboratory strains.

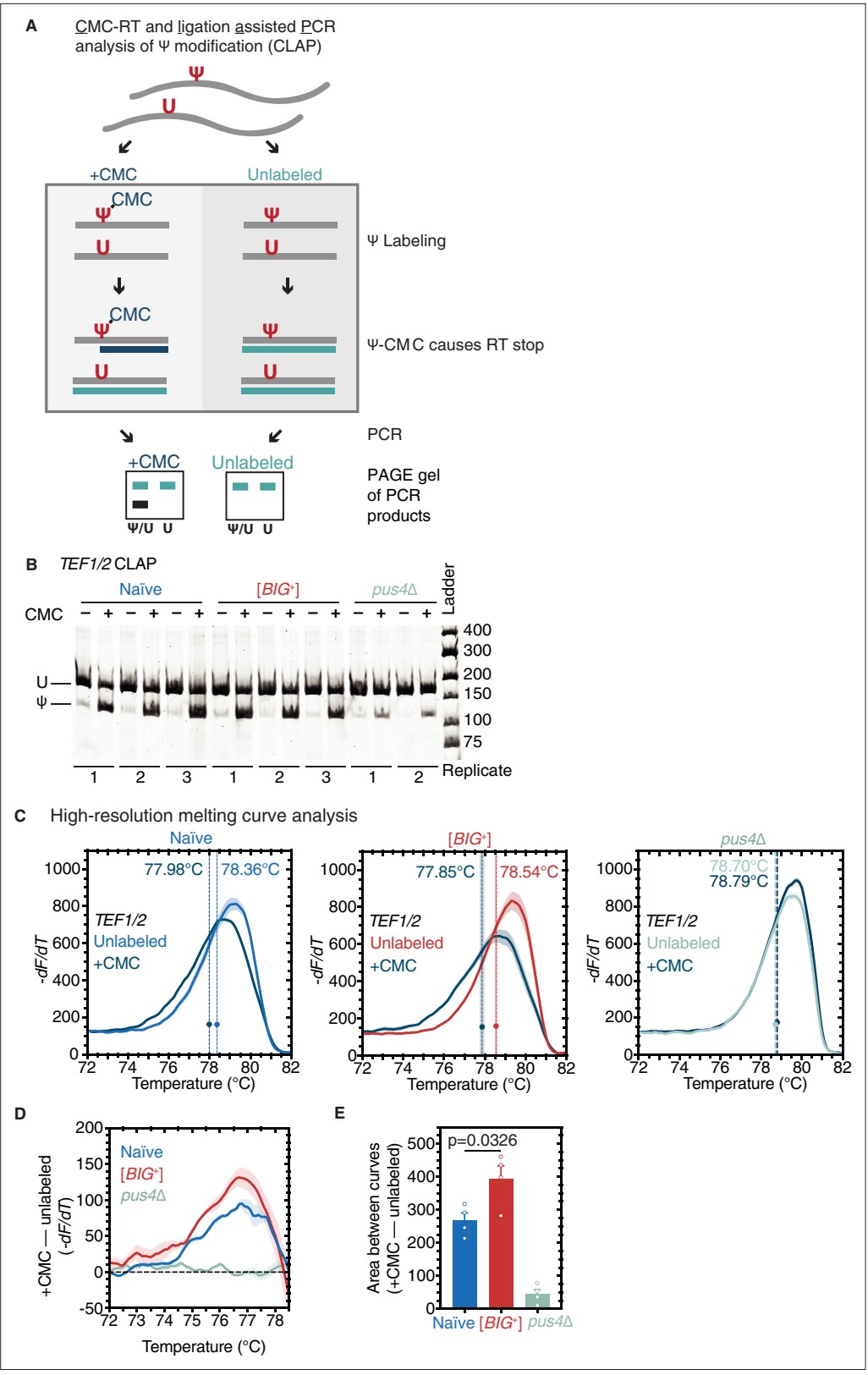

**Figure 7.** Pseudouridylation of mRNA in [*BIG*+] cells. (**A**) CMC-RT and *ligation-assisted PCR* analysis of Ψ modification (CLAP) (*Zhang et al., 2019*). (**B**) Native PAGE showing the CLAP result for *TEF1/TEF2* mRNA from multiple replicates of naïve (3), [*BIG*+] (3), or *pus4Δ* (2). Shorter bands indicating pseudouridylation that appear in CMC-unlabeled or *pus4Δ* samples may reflect either background signal from spontaneous cleavage near

*Figure 7 continued on next page*

*Figure 7 continued*

target site or Pus4p-independent modification of this site. (**C**) High-resolution melting curve analysis shows that U239 is pseudouridylated in *TEF1/TEF2* mRNA in both naïve and [*BIG*⁺] cells but not in cells that do not contain Pus4p. Left panel: naïve samples CMC-labeled (dark blue) or -unlabeled (light blue). Middle panel: [*BIG*⁺] samples CMC-labeled (dark blue) or -unlabeled (red). Right panel: *pus4Δ* samples CMC-labeled (dark blue) or -unlabeled (turquoise). Dots mark the geometric center of four replicates, bisected by a dashed line with shaded area representing the standard error of the mean. The melting temperature (Tm) of this point is also displayed. Solid lines representing melting curves are the mean of four replicates, with shaded areas representing the standard error of the mean. (**D**) The difference in melting temperature behavior (*df/dT*) between CMC-labeled and CMC-unlabeled *TEF1/TEF2* mRNA amplicons is larger in [*BIG*⁺] cells than in naïve cells, suggesting higher levels of pseudouridylation of U239 in [*BIG*⁺] cells. Solid line represents the mean of four replicates, with shaded areas showing standard error of the mean. (**E**) The difference in area between the melting curves of CMC-labeled and CMC-unlabeled *TEF1/TEF2* mRNA amplicons is greater in [*BIG*⁺] cells than in naïve cells, suggesting higher levels of pseudouridylation of U239 in [*BIG*⁺] cells. Bars represent the mean of four replicates, error bars indicate standard deviation, p=0.0326, unpaired t-test.

The online version of this article includes the following figure supplement(s) for figure 7:

**Figure supplement 1.** Pseudouridylation of mRNA in [*BIG*⁺] cells.

Second, we examined the localization of Pus4 in BC187 and its Hsp70 cured derivative. We transformed uncured and cured strains with a plasmid expressing GFP-tagged Pus4 protein and imaged cells using epifluorescence microscopy. In uncured cells, we observed Pus4 in a fragmented network throughout the cytoplasm (**Figure 8C**), similar to what we observed in laboratory [*BIG*⁺] cells (**Figure 4G**). In contrast, in cured BC187 cells, the signal was more diffuse.

Finally, we investigated the relationship between Pus4 and cell size in BC187. The original [*BIG*⁺] trait identified in laboratory yeast strains (this work and **Chakrabortee et al., 2016a**) was induced after transient overexpression of Pus4 protein. Therefore, we tested whether transient overexpression of Pus4 protein could re-establish the large-cell phenotype in BC187 derivatives that had been subject to curing by transient Hsp70 inhibition. Indeed, cured BC187 cells treated in this way became large (**Figure 8D**). By contrast, transient overexpression of Pus4 did not increase the size of the already-large BC187 natural isolate (**Figure 8—figure supplement 1**). Thus, re-initiation of this epigenetic trait could be accomplished with transient Pus4 overexpression alone, thereby bearing strong similarities to [*BIG*⁺] in common laboratory strains and establishing Pus4 as a driver of epigenetic cell size control in this model eukaryote. Together, these observations suggest that epigenetic and potentially Pus4 prion-mediated control of mechanisms like [*BIG*⁺] that we have described in the laboratory is also likely present in nature.

## Discussion

Epigenetic inheritance is most commonly thought to be driven by enzymes that modify chromatin and DNA. Here we show that an enzyme that catalyzes the epigenetic modification of RNA can itself be controlled by an extrachromosomal epigenetic process: a self-templating protein conformation that persists over long biological timescales. We found that prion cells not only maintain pseudouridylation activity but that activity can be increased, even without any detectable increase in *PUS4* expression. Future studies should address whether Pus4-dependent modification of *TEF1/TEF2* mRNA plays a role in the phenotypes of [*BIG*⁺] as this mRNA, which is the major non-tRNA substrate of Pus4, encodes a central elongation factor that binds to and escorts tRNAs to the ribosome. The modest changes we observed in the relative abundance of a few dozen mRNAs in actively growing cells do not appear to drive these growth or aging phenotypes based on their known functions.

Instead, we found that changes to the translational control of numerous genes are logically connected to these phenotypes. These include genes that affect cell size, chronological lifespan and replicative lifespan, translation control and ribosome biogenesis, and differential sensitivity to rapamycin (**Supplementary file 3**). Translation is a rate-limiting step for growth in many organisms (**Polymenis and Schmidt, 1997**; **Sonenberg, 1993**) and is often activated in human cancers (**Sonenberg and Hinnebusch, 2009**). Other prions in yeast also affect translation, including [*PSI*⁺] and [*MOD*⁺] (**Baudin-Baillieu et al., 2014**; **Suzuki et al., 2012**). However, in contrast to [*BIG*⁺], they lead to losses of their underlying protein activities, impairing translation and, in turn, growth in many conditions

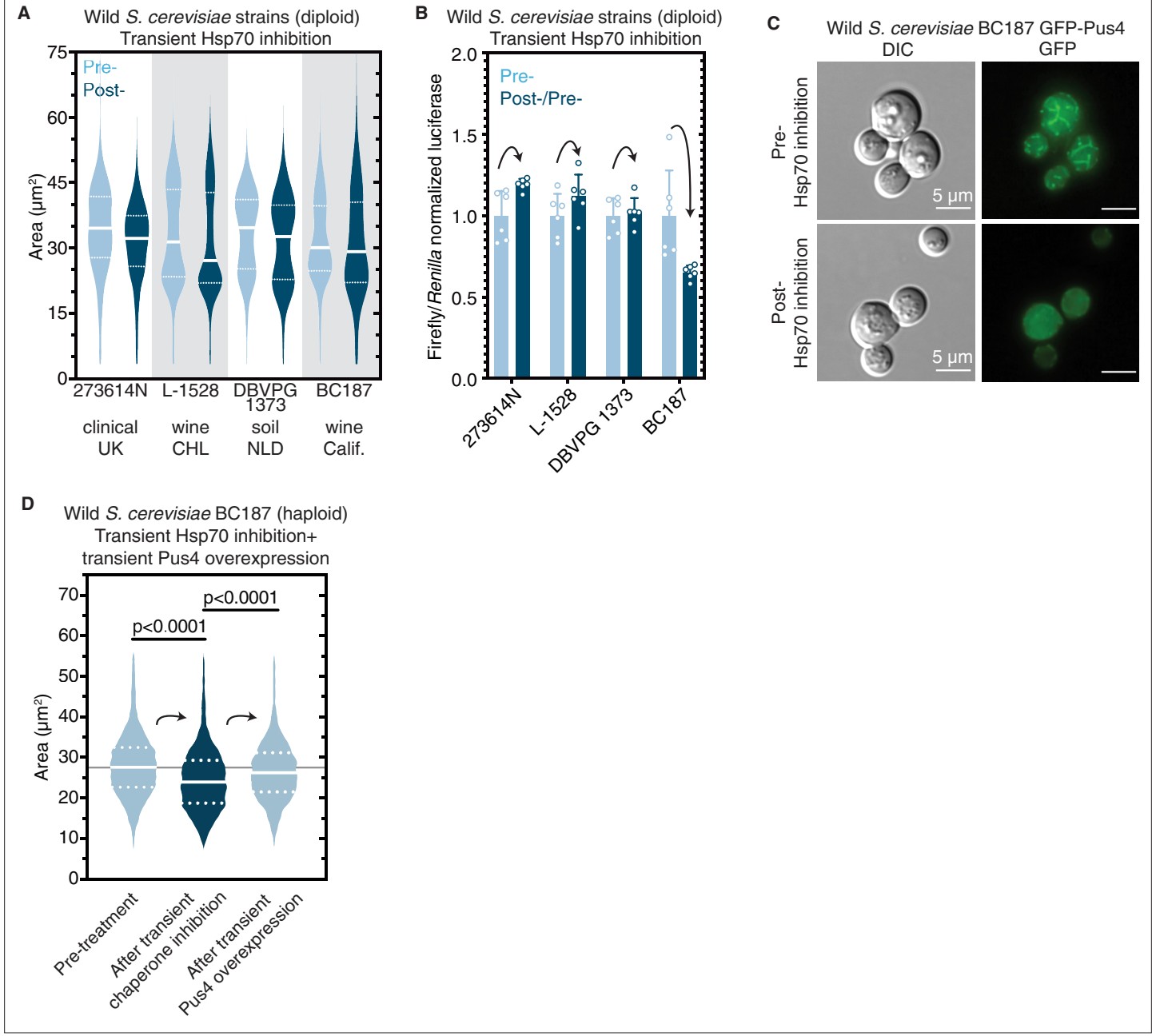

**Figure 8.** Epigenetic control of [*BIG*⁺]-like phenotypes in wild yeasts. (**A**) Transient inhibition of Hsp70 in diploid wild yeast strains from different niches around the globe leads to a permanent reduction in cell size. Violin plots show all data from thousands of cells from three biological replicates of each strain, light blue are cells before Hsp70 inhibition ('pre-'), dark blue are cells after transient Hsp70 inhibition and recovery ('post-'). The solid white line bisecting each distribution indicates mean; dotted lines indicate upper and lower quartiles. 273614 N: clinical isolate from the United Kingdom, pre- vs. post- p<0.0001. L-1528: wine isolate from Chile, pre- vs. post- p<0.0001. DBVPG 1373: soil isolate from the Netherlands, pre- vs. post- p<0.0001. BC187: wine isolate from California, pre- vs. post- p<0.0001. Kolmogorov–Smirnov test for all. (**B**) Transient inhibition of Hsp70 in wild yeast strains leads to permanent changes in protein synthesis capacity. Firefly reporter contains the 'normal' suite of codons, which is normalized to internal control *Renilla* luciferase. Bars represent the mean of normalized luciferase values for six biological replicates, error bars are standard deviation. Light blue 'pre-' are cells prior to Hsp70 inhibition, dark blue 'post-' are cells after transient Hsp70 inhibition and recovery. A value above 1.0 for the dark blue bars indicates that the normalized translation capacity has increased after prion curing, a value below 1.0 indicates that capacity has decreased. 273614 N p=0.0134, L-1528 p=0.1397, DBVPG 1373 p=0.7428, BC187 p=0.0125, unpaired t-test for all. (**C**) Transient inhibition of Hsp70 in BC187 wine isolate leads to permanent changes in Pus4p expression pattern, suggesting that its conformation may also be epigenetically regulated in wild strains. Prior to inhibition (top), cells show a distinct fragmented network of Pus4 expression; after transient inhibition and recovery (bottom), the expression pattern becomes much more diffuse. (**D**) Transient inhibition of Hsp70 in haploid wild BC187 yeast strains leads to a permanent reduction in cell size. After this regime, if

*Figure 8 continued on next page*

*Figure 8 continued*

Pus4p alone is transiently overexpressed, cells then permanently regain their original larger cell size. Violin plots show all data from thousands of cells from three biological replicates of each strain. The solid white line bisecting each distribution indicates mean; dotted lines indicate upper and lower quartiles. Kolmogorov–Smirnov tests.

The online version of this article includes the following figure supplement(s) for figure 8:

**Figure supplement 1.** Transient Pus4 overexpression does not permanently increase the size of uncured BC187 cells.

(*Baudin-Baillieu et al., 2014*; *Cox, 1965*; *Suzuki et al., 2012*). In contrast, in [*BIG*+] cells, translation appears to be amplified—it is faster, leading to higher expression levels of many proteins. We are unaware of an example in the literature of a mutation that enhances translation under nutrient-replete conditions.

More investigation is needed to determine the precise mechanism by which translation is altered. Increased polysome density can be associated with increased translation (more protein produced per mRNA) or decreased translation (due to defects in elongation or termination leading to accumulation of ribosomes). However, multiple lines of evidence lead us to favor the former interpretation. First, [*BIG*+] cells are resistant to the translation elongation inhibitor cycloheximide. Second, their rates of bulk translation as measured by [$^{35}$S]-methionine incorporation are also higher than naïve cells. General defects in translation elongation or termination, by contrast, would be expected to reduce this activity. Our luciferase reporter data shows an improved translation of mRNAs containing rare codons, which slow translation elongation. Finally, when measuring the proteome-wide effects on translation using our GFP screen, not only did we observe logical connections between the gene functions of altered proteins and the observed [*BIG*+] growth phenotypes, but it was the proteins whose expression increased that contained a feature consistent with enhanced elongation of their mRNAs: their ORF 5′ ends had a lower CAI. Collectively, these data suggest an increase in translation elongation efficiency of many mRNAs as a parsimonious model.

The various possible mechanisms need not be mutually exclusive—some mRNAs in [*BIG*+] cells may indeed be translated faster to produce more protein without any defects and perhaps exhibit elevated rates of elongation and termination. In contrast, other mRNAs in the same cells may be subject to deficiencies at any of these steps. Examining protein synthesis at higher resolution, using methods like ribosome profiling, and in response to perturbations to ribosome-associated quality control pathways (*Brandman and Hegde, 2016*), may reveal the landscape of changes to protein synthesis in [*BIG*+] cells in greater detail.

Translation is also coupled to cell size, proliferation, and lifespan (*Ecker and Schaechter, 1963*; *Kaeberlein and Kennedy, 2007*; *Lloyd, 2013*; *Steffen and Dillin, 2016*; *Tanenbaum et al., 2015*). Cell size, which is determined in large part by the activity of the TOR pathway (*Fingar et al., 2002*; *Zhang et al., 2000*), has been inversely correlated with lifespan (*Anzi et al., 2018*; *He et al., 2014*; *Yang et al., 2011*), and older cells are larger (*Egilmez et al., 1990*). Moreover, molecules that extend lifespan, such as rapamycin, also influence cell size and proliferation by restricting the cell's translation capacity (*Beretta et al., 1996*; *Terada et al., 1994*). Here we describe an epigenetic paradigm that links all of these fundamental cellular properties: translation, cell size, proliferation, and lifespan. In future studies, we plan to explore to what extent the effect on lifespan may serve as material for selection to favor or disfavor [*BIG*+]. At present, we favor a hypothesis in which the aging defect of [*BIG*+] cells is due at least in part to pleiotropic consequences of their increased proliferation and translation capacity. Although these features have already been linked to aging in genetic studies, we note that such theories of antagonistic pleiotropy in aging are not without controversy (*Hughes and Reynolds, 2005*).

The replicative lifespan of wild budding yeast strains varies widely (*Kaya et al., 2015*) and has been associated with changes in oxidative phosphorylation, respiration, and differences in metabolite biosynthesis. Genetic screening has also offered some insight into the genetic basis of this variability (*McCormick et al., 2015*). Lastly, genetic mapping efforts have identified polymorphisms underlying natural variation in chronological lifespan variation (*Kwan et al., 2011*). The genetic architecture of natural lifespan, both chronological and replicative, remains obscure, however. Our data further demonstrate that it can be subject to strong epigenetic control.

We note that prior studies have characterized genetic links between cell size and lifespan in yeast— mutants that make cells larger tend to age faster, and older cells tend to be larger than younger cells

(*Neurohr et al., 2019*; *Yang et al., 2011*; *Zadrag-Tecza et al., 2009*). [*BIG*+] provides an epigenetic mechanism to alter these relationships heritably. Exerting epigenetic, rather than genetic, control over basic cell growth behaviors could be valuable in the face of fluctuations between nutrient-replete and nutrient-poor conditions. Theory predicts that such mechanisms can have substantial adaptive value when the frequency of environmental fluctuations is rare relative to the frequency of phenotypic change (*King and Masel, 2007*). In agreement with these inferences, our modeling quantitatively described the long-run selective advantage of this '*live fast, die young*' prion state that we measured, illustrating the importance of considering not only steady-state phenotypes but also the ecological context in which prion states are expressed. Of particular relevance to this point is our data demonstrating that transient perturbation of Hsp70 activity can eliminate [*BIG*+]. Conditions in which the prion confers a growth benefit match those that promote its propagation. Conversely, conditions in which the prion is detrimental, shortening lifespan, are also known to reduce chaperone expression (*Janssens et al., 2015*; *Werner-Washburne et al., 1989*; *Werner-Washburne and Craig, 1989*) and could thereby cure the prion. The strong dependence of [*BIG*+] on chaperones therefore means that it is a natural epigenetic sensor of its environment.

Our data from wild yeast isolates demonstrates that cell size and protein synthesis can also be under epigenetic control in natural isolates and is influenced by Pus4. Exploring whether a [*BIG*+]-like epigenetic mechanism that promotes proliferation in nutrient-replete conditions is conserved in metazoans is a major goal for the future. Indeed, cancer cells also experience frequent oscillations in their environments: as tumors grow and metastasize, cells are exposed to shifting gradients of oxygen and glucose that influence their proliferation rate (*Martinez-Outschoorn et al., 2017*; *Schito and Semenza, 2016*). Regulation of cell physiology in these situations is best understood at the level of transcriptional changes due to the relative ease of profiling them. However, multiple lines of evidence suggest that translational hyperactivation can also fuel pathological proliferation (*Robichaud et al., 2019*). Our discovery of a protein-based mechanism that changes cell growth via engagement of an altered translational program argues for greater investigation into the epigenetic control of post-transcriptional processes in both normal biology and disease.

## Materials and methods

### Key resources table

| Reagent type (species) or resource | Designation | Source or reference | Identifiers | Additional information |
|---|---|---|---|---|
| Strain, strain background (*Saccharomyces cerevisiae*) | | | | See *Supplementary file 4* |
| Antibody | Anti-FLAG M2 mouse monoclonal antibody | Sigma | RRID:AB_262044 | (1:1000) |
| Antibody | Goat anti-mouse IgG peroxidase | Sigma | RRID:AB_257993 | (1:10,000) |
| Antibody | Mouse anti-PGK1 monoclonal antibody | Invitrogen | RRID:AB_2532235 | (1:1000) |
| Recombinant DNA reagent | | | | See *Supplementary file 4* |
| Sequence-based reagent | | | | See *Supplementary file 4* |
| Commercial assay or kit | Dual-Luciferase Reporter Assay | Promega | Catalog# E1910 | |
| Commercial assay or kit | ERCC ExFold RNA Spike-In Mixes | Invitrogen | Catalog# 4456739 | |
| Chemical compound, drug | Cycloheximide | Sigma | Catalog# C1988 | CAS# 66-81-9 |
| Chemical compound, drug | Radicicol | Research Products International | Catalog#R20020-0.001 | CAS# 12772-57-5 |
| Chemical compound, drug | Rapamycin | LC Laboratories | Catalog# R-5000 | CAS# 53123-88-9 |
| Chemical compound, drug | Canavanine | Sigma | Catalog# C9758 | CAS# 2219-31-0 |
| Chemical compound, drug | 5-Fluoroorotic acid monohydrate (5-FOA) | GoldBio | Catalog# F-230–2.5 | CAS# 220141-70-8 |
| Chemical compound, drug | Guanidine hydrochloride | Sigma | Catalog# G3272 | CAS# 50-01-1 |

*Continued on next page*

*Continued*

| Reagent type (species) or resource | Designation | Source or reference | Identifiers | Additional information |
|---|---|---|---|---|
| Chemical compound, drug | [$^{35}$ S]-methionine | Perkin Elmer | Catalog# NEG709A500UC | |
| Chemical compound, drug | *N*-Cyclohexyl-*N'*-(2-morpholinoethyl)carbodiimide methyl-*p*-toluenesulfonate (CMC) | Sigma | Catalog# C106402 | CAS# 2491-17-0 |
| Software, algorithm | ImageJ software | ImageJ (http://imagej.nih.gov/ij/) | RRID:SCR_003070 | |
| Software, algorithm | GraphPad Prism software | GraphPad Prism (https://graphpad.com) | RRID:SCR_002798 | |
| Software, algorithm | Model | https://github.com/cjakobson/liveFastDieYoung | | |

## Model formulation

To model the fitness of an epigenetic element for which growth in nutrient-replete conditions is improved and survival in starvation conditions is worsened, we define the following growth equations in nutrient-replete conditions:

$$x_0 = x_{00}e^{\mu_0 t}$$

$$x_1 = x_{10}e^{\mu_1 t}$$

where $x_0$ represents the population of naïve cells and $x_1$ represents the population of [*BIG$^+$*] cells. $\mu_0$ and $\mu_1$ are the growth rates of naïve and [*BIG$^+$*] cells, respectively. We neglect the lag and stationary phases of growth, as the ratio between populations does not change during this time.

Likewise, in starvation,

$$x_0 = x_{00}e^{-\delta_0 t}$$

$$x_1 = x_{10}e^{-\delta_1 t}$$

where $\delta_0$ and $\delta_1$ are the death rates of naïve and [*BIG$^+$*] cells, respectively.

We can furthermore define the times of nutrient repletion and starvation as $\tau_1$ and $\tau_2$, respectively. Thus, for each cycle of nutrient/starvation, we can define the following recursion relations:

$$x_{0,i+1} = x_{0,i}e^{\mu_0 \tau_1}$$

$$x_{1,i+1} = x_{1,i}e^{\mu_1 \tau_1}$$

$$x_{0,i+2} = x_{0,i+1}e^{-\delta_0 \tau_2}$$

$$x_{1,i+2} = x_{1,i+1}e^{-\delta_1 \tau_2}$$

And we can write an analytical expression for the ratio of populations after one repletion/starvation cycle:

$$x_0\left(t + \tau_1 + \tau_2\right) = x_0\left(t\right)e^{\mu_0 \tau_1 - \delta_0 \tau_2}$$

$$x_1\left(t + \tau_1 + \tau_2\right) = x_1\left(t\right)e^{\mu_1 \tau_1 - \delta_1 \tau_2}$$

$$\frac{x_1}{x_0}\left(t + \tau_1 + \tau_2\right) = \frac{x_1(t)}{x_0(t)}e^{\left(\mu_1 \tau_1 - \delta_1 \tau_2\right) - \left(\mu_0 \tau_1 - \delta_0 \tau_2\right)}$$

For the purposes of the model, we consider only the ratio between populations in the two epigenetic states (neglecting the total carrying capacity of the environment). The above defines a recursion relation from which we can predict the ratio of naïve and [*BIG*⁺] cells after N cycles of nutrient repletion and starvation.

## Parameter estimation and prediction of competitive fitness

The free parameters defining the growth advantage and starvation disadvantage attributable to the epigenetic element in the model above ($\mu_0$ and $\mu_1$; $\delta_0$ and $\delta_1$) were determined by Monte Carlo sampling of independent measurements of the growth and death rates (*Figure 2D and E*). To generate the ensemble of competitive fitness predictions shown in *Figure 2C*, we randomly sampled growth and death rates for naïve and [*BIG*⁺] cells according to their experimentally determined distributions. The sampling was conducted 1000 times, and the prediction shown in *Figure 2F* is the median and 95% confidence interval of the resulting ensemble of model predictions.

## Code and data availability

All MATLAB code and data required to generate the model predictions are available at http://github.com/cjakobson/liveFastDieYoung (copy archieved at swh:1:rev:d86455c2e53f50644862690e22bfdc-d080abdf63, *Jakobson, 2021*). Code to generate the figures is available upon reasonable request to jarosz@stanford.edu.

## Bacterial strain growth

Bacteria strains (for plasmid propagation) were cultured on LB agar or liquid (Research Products International (RPI), Mount Prospect, IL).

## Yeast strains

Yeast strains were cultured on either YPD agar or liquid (RPI) or SD-Ura (Sunrise Scientific, Knoxville, TN) unless otherwise indicated. Strains were stored as glycerol stocks (25% glycerol [Amersco, Solon, OH] in appropriate media) at –80 °C and revived on YPD or amino acid dropout media before testing. Yeast were grown in YPD at 30 °C on a TC-7 roller drum wheel (New Brunswick) unless indicated otherwise. Yeast transformations were performed with a standard lithium–acetate protocol (*Gietz et al., 1992*). The *pus4Δ* strain was sourced from the BY4741 MATa haploid knockout library (GE Dharmacon, Lafayette, CO).

## Strain constructions

Most diploids were constructed by crossing indicated BY4741 haploids to the BY4742 parental strain (ATCC, Manassas, VA) by mixing a bead of cells of each strain (from a single colony) together on a YPD plate and growing overnight at 30 °C. A small globule of this cell mixture was then restreaked to single colonies on SD-Lys-Met agar plates to select for diploids.

Diploids constructed for the experiments presented in *Figure 4G* and *Figure 6—figure supplement 1A* were made by crossing [*BIG*⁺] or naïve haploids to the PCR-verified seamless GFP-Pus4 or seamless GFP-Par32 haploid strains, respectively. Due to overlapping auxotrophic makers in each parent strain, diploids could not be selected on dropout plates from cell mixtures. Instead, pools of the mated cells were grown in several successive competitions to allow diploids to outcompete haploid parents. Diploids were then isolated from single colonies, and verified using either microscopy to measure cell size (GFP-Pus4), which is significantly larger than either haploid parent, or PCR amplification of the GFP tag/ORF junction (GFP-Par32), which yielded two distinct bands matching the expected sizes of the wild-type allele and the tagged allele, in contrast to the haploid parental strains that yielded only one or the other band sizes. These isolated diploids were then imaged as described below.

Sporulations were performed inoculating single diploid colonies in Pre-SPO liquid media (YPD with 6% glucose) for 2 days at room temperature on a roller drum wheel. Cells were then pelleted and washed twice in SPO media (1% potassium acetate [Sigma, St. Louis, MO], 320 mg CSM-Met powder [Sunrise Scientific], 20 mg methionine [Sigma] per liter), and then diluted 10-fold into 3 mL cultures of SPO. These cultures were incubated on a rotary wheel for 1 week at room temperature, before dissecting tetrads on a Singer Instruments MSM400 (Somerset, England).

The 3XFLAG-Pus4 strains were constructed using PCR combined with the 'Delitto Perfetto' method (**Storici and Resnick, 2006**). Canavanine mutants were constructed by pelleting 500 µL of a saturated YPD liquid culture, resuspending it in 100 µL YPD and plating this on SD-Arg agar plates containing 60 µg/mL canavanine (Sigma), and growing for 2 days at 30 °C. Single canavanine-resistant colonies were picked and re-tested for resistance before further testing.

Cytoductions were performed as described in **Chakrabortee et al., 2016a**. A BY4742 strain with a defective *KAR1* allele (*kar1-15*) was created as an initial recipient for cytoplasmic transfer. This allele prevents nuclear fusion during mating while permitting cytoplasmic transfer. The strain carries auxotrophic markers distinct from those in the putative [*BIG*+] or naïve donor strains, and was also converted to petite with growth on ethidium bromide (strains were grown in YPD with 25   µg/mL ethidium bromide for approximately two-dozen generations before testing for growth on YP-glycerol). This allowed cytoplasmic transfer to be scored through the restoration of mitochondrial respiration, while selecting for auxotrophic markers unique to the recipient strain. The recipient and donor strains were mixed together on YPD-agar and grown overnight, followed by selection of heterokaryons and resulting haploid cytoductants on dropout media (selecting for the BY4742 recipient strain markers) containing glycerol as a carbon source. One more round of selection was used while replica-plating onto a dual-selection agar plate (SD-Lys-Met) to confirm that the colonies were not diploids. One additional round of propagation on a non-selective plate was performed before doing "reverse cytoductions,' which were performed in the same way except selecting for BY4741 auxotrophy in recipient naïve strain. In the reverse cytoductions, the donors were naïve or putative-[*BIG*+] BY4742 *kar1-15* cytoductants from the first round, and the recipients were wild-type or *pus4Δ* naïve BY4741 petite cells.

Transient *PUS4* deletion experiment strains were made by the Delitto Perfetto method (**Storici and Resnick, 2006**). After deletion of *PUS4*, strains were propagated for ~75 generations before phenotyping. The *PUS4* gene was re-introduced by homologous recombination.

Transient overexpression of Pus4 in haploid BC187 yeast was performed using PDJ541 (CEN URA3 GAL::PUS4) (**Supplementary file 4**). Three independent transformants of BC187 (previously cured by transient Hsp70 inhibition) bearing this plasmid were grown in SGalactose-Ura for 48 hr at 30 °C. Overexpression was stopped by outgrowth in non-selective YPD medium for 48 hr. Strains lacking the overexpression plasmid were isolated by plating to single colonies on media containing 5-FOA. Uracil auxotrophy was confirmed before subsequent measurements of cell size.

## Growth assays

Biological replicates of each yeast strain were pre-grown in rich media (YPD). We then diluted these saturated cultures 1:20 in sterile water and then inoculated 3 µL into 96-well humidified plates (Nunc Edge Plates [Thermo Scientific, Waltham, MA]) with 150 µL of YPD or SD-CSM per well. Cycloheximide (Sigma) was added to growth media at either 0.01 µg/mL or 0.05 µg/mL, as indicated. Rapamycin (LC Laboratories, Woburn, MA) was added to growth media at 10 µM. Cell growth was monitored with continuous measurements of $OD_{600}$ (approximately every 10 min) at 30 °C over 96 hr using BioTek Eon or Synergy H1 microplate readers (Winooski, VT). Timepoints plotted in bar graphs correspond to the maximum proliferation rates calculated from growth data.

## Measurement of chronological lifespan

For each strain, four single colonies were picked from freshly streaked YPD plate and grown in 5 mL of pre-sporulation media for 3 days on a roller drum wheel (New Brunswick Scientific, Edison, NJ) at 30 °C. Cultures were then pelleted and washed once with SPO media, and resuspended in 5 mL of SPO media, and placed back on the roller drum wheel at 30 °C. On days indicated, a dilution was made of each replicate to achieve dozens to hundreds of colonies on a YPD plate, which were then counted using a colony counter (Synbiosos Acolyte, Frederick, MD). Dilution was ~100,000× at early stages of the experiment and was later empirically determined due to significant cell death. We note that aging the cells in SPO did not lead to significant acidification (pH of old cultures was found to be >5), as has been reported for cells aged in YPD, which contains high levels of glucose that upon metabolism leads to the secretion of organic acids (**Murakami et al., 2011**).

## Measurement of replicative lifespan

RLS was assessed using the standard method of isolating virgin cells on agar YPD (2% glucose) plates, and then separating their daughter cells at each cell division by micromanipulation and counting the total number of daughters produced by each mother cell (*Steffen et al., 2009*; *Wasko et al., 2013*). Strains were streaked from glycerol stocks onto YPD plates and allowed to grow at 30 °C until individual colonies could be selected for each strain. Colonies were lightly patched onto fresh YPD overnight and 20 cells were isolated by microdissection from each patch. These cells were incubated at 30 °C for ~2 hr until they had formed daughter cells, at which time individual virgin daughter cells were selected and arrayed as previously described (*Steffen et al., 2009*) for lifespan analysis. From these, daughter cells were removed by microdissection and counted approximately every 2 hr during the day. Plates were maintained at 30 °C during the day and placed at 4 °C overnight. At least four independent replicates (arising from different colonies) of 20 individual mother cells each were measured for each strain.

## Curing

Three regimes of chaperone inhibition were tested: (1) transient exposure to a dominant-negative version of Hsp70 (Ssa1) to inhibit its activity (*Chakrabortee et al., 2016a*; *Jarosz et al., 2014*); (2) transient exposure to radicicol (RPI) to inhibit Hsp90 activity (*Chakrabortee et al., 2016a*); and (3) transient exposure to guanidinium hydrochloride (Sigma) to inhibit Hsp104 activity (*Ferreira et al., 2001*).

Regime 1 was performed by transforming cells with a plasmid, PDJ169, harboring a dominant-negative version of Hsp70 (Ssa1) as described previously (*Chakrabortee et al., 2016a*; *Jarosz et al., 2014*; *Lagaudriere-Gesbert et al., 2002*). Transformants were picked and restreaked by hand or replica-pinned using a Singer HDA robot a total of 12 times on SD-Ura to promote Ssa1$^{DN}$ expression. (Anecdotally, we note that prion phenotypes are frequently cured with fewer than 12 restreaks, but for reasons of technical throughput, 12 were used in this experiment.) Then plasmids were eliminated by plating on media containing 5-fluoroortic acid (SD-Ura +0.1 % 5-FOA + 50 µg/mL uracil) and plasmid loss was verified by replating on SD-Ura. Colonies were then tested for the elimination of prion phenotypes. Tested strains were compared to control strains that were restreaked in parallel on SD-CSM plates.

Regime 2 was performed by replica-pinning cells six subsequent times on YPD agar plates containing 5 µM radicicol, using a Singer HDA robot. After replatings, cells were plated back onto YPD two subsequent times to facilitate recovery before being tested for the elimination of prion phenotypes. Tested strains were compared to control strains that were replica-pinned in parallel on YPD plates.

Regime 3 was performed like regime 2 but with SD-CSM plates containing 0.5 g/L guanidinium hydrochloride. Tested strains were compared to control strains that were replica-pinned in parallel on SD-CSM plates.

Wild yeast strains were cured by transforming a uracil-selectable 2micron plasmid (PDJ1222) encoding the aforementioned dominant-negative version of Hsp70 (Ssa1), under control of a constitutive promoter (*pGPD*). Transformants were passaged on selective media five times to allow the growth of single colonies. Transformants were then passaged three times on non-selective media (YPD) to permit plasmid loss, which was confirmed by the lack of growth on selective media (SD-Ura).

## Strain competitions

Single colonies were used to inoculate 5 mL YPD cultures that were grown for 3 days on a roller drum wheel at 30 °C. Cells were diluted 1000-fold and then mixed in equal volumes to form 50:50 mixtures of either of the following: naïve Can$^S$ and [*BIG$^+$*] Can$^R$; or naïve Can$^R$ and [*BIG$^+$*] Can$^S$. Before mixing cells, saturated cultures were measured to have near equal cell densities, and 'time zero' measurement was made by plating the initial cell mixture and counting the number of canavanine-resistant colony-forming units (CFUs) relative to total CFUs. These initial strain mixtures were then grown for 2 days on a roller drum wheel at 30 °C, after which cells were diluted 50,000-fold or 25,000-fold and plated on YPD or canavanine plates, respectively. These plates were grown at 30 °C for 2 days before counting colonies. The liquid cultures were diluted 1:1000 in 5 mL of fresh YPD, and this process was repeated nine more times. Swapping of canavanine resistance between naïve and [*BIG$^+$*] was done

to correct for the potential of canavanine resistance to influence cell growth. However, in our experiments the differences were negligible. The numbers of canavanine-resistant and total colonies were compared relative to a number of cells plated to determine the number of naïve or [*BIG*⁺] colonies arising at each timepoint.

## Microscopy and cell size measurements

Most microscopy was performed using a Leica inverted epifluorescence microscope (DMI6000B) with a Hamamatsu Orca 4.0 camera. Cells were imaged after 3 days of growth in 5 mL YPD at 30 °C. Saturated cultures were diluted 10-fold with 1 X PBS and briefly sonicated to break up cell clumps. Differential interference contrast (DIC) images were taken at 20 ms exposure time using a 63× /1.40 oil objective. Cell area was calculated using CellProfiler (3.1.5) image analysis software (http://www.cellprofiler.org; *Carpenter et al., 2006*). The large cell threshold was set at one standard deviation above the cell area mean of the naïve cells.

GFP microscopy data presented in *Figure 4G* were obtained from cells were grown in YPD for 24 hr. Aliquots of 1 mL were spun down and resuspended in 200 μL 1 X PBS, 2 μL of which were prepared into an agar pad. Agar pad made with 2 % agar and PBS. Fluorescent images were taken at 400 ms exposure time and a Z-stack consisting of 25 steps for a total length of 4.83 μm in each channel using a 100× /1.40 oil objective.

Data presented in *Figure 8C* were imaged similarly to *Figure 4G*, except that cells were grown in SC-Ura media for retention of the GFP-Pus4 expression plasmid. This plasmid (*Supplementary file 4*) overexpresses Pus4 protein via a strong promoter driving the expression of GFP-Pus4 constitutively on a centromeric plasmid.

Data presented in *Figure 6—figure supplement 1A* were grown and imaged similarly to above, except cells were imaged in 1 X PBS using a GE DeltaVision Ultra microscope (Boston, MA).

Cell size experiments using protein synthesis inhibitors (*Figure 6A–D*): conditions were the same as above, with the following differences. Single colonies were inoculated into YPD, YPD + cycloheximide (0.05 μg/mL), or YPD + rapamycin (10 μM) and grown for 4 days before imaging. (Very similar results were observed after 3 days of growth.) The following day, cultures were diluted into liquid YPD and grown for 3 days, after which they were restreaked once onto YPD agar. Single colonies were then used to inoculate liquid YPD cultures, grown for 3 days before imaging to test for the reappearance of the large cell size phenotype.

## Microscopy image processing

ImageJ version 2.0.0-rc-69/1.52 p, Build 269a0ad53f.

For GFP-Pus4 in laboratory *S. cerevisiae* and wild *S. cerevisiae* isolate BC187 (*Figures 4G and 8C*): (1) using full-sized images, created a maximum projection of 25-step Z-stack; (2) in ImageJ, set 50 × 50 pixel square in the area between cells and measure average intensity; (3) subtracted this intensity from the total image using the 'Math' function; (4) added 5 μm scale bar calculated from the original voxel size of 0.065 μm; and (5) enlarged Photos 6X using Topaz Gigapixel AI software version 5.3.2 (Topaz Labs).

For GFP-Par32 microscopy (*Figure 6—figure supplement 1A*), analysis was performed using ImageJ (imagej.nih.gov). All images were adjusted to a uniform contrast and illumination corrected using standard tools in ImageJ. The background was calculated using a morphological opening with a disc of radius 75 pixels, such that the structuring element was larger than cells in the foreground. In these background-corrected images, we quantified the ratio of membrane to cytoplasmic Par32-GFP within each cell by measuring the average intensity for a region at the plasma membrane and the average intensity for an equivalently sized region approximately 0.2 μm from the plasma membrane. For each cell, the 'Fraction membrane Par32 signal' was calculated by dividing the average Par32-GFP intensity in the membrane region by the average Par32 intensity in the cytoplasmic region.

## Luciferase assays

Strains were transformed with PDJ512 and PDJ513. To maintain the plasmids, we grew these cells in a synthetic complete medium containing nutrient levels between those in SD and YPD formulations. Four independent transformants for each sample were grown for 1 day in 150 μL SC-Ura (Sunrise Scientific) per well in 96-well plates at 30 °C. Saturated cultures were then diluted 15 × into fresh

media in a new 96-well plate and grown until cultures reach $OD_{600}$ 0.6, as determined by a BioTek Eon plate reader. 20 µL of each culture was added using a multichannel pipette into a white flat-bottom 96-well microplate (E&K Scientific, Santa Clara, CA) already containing 20 µL of room temperature 1 X Passive Lysis Buffer from the Dual-Luciferase Reporter Assay System (Promega, Madison, WI). Cultures were then lysed by shaking at 300 rpm for 25 min at room temperature. *Renilla* and firefly luciferase activity was measured using 75 µL injection volumes and otherwise default settings on a Veritas luminometer (Turner Biosystems). pTH726-CEN-RLuc/minCFLuc (PDJ512) and pTH727-CEN-RLuc/staCFLuc (PDJ513) were gifts from Tobias von der Haar (University of Kent; Addgene plasmids #38210 and #38211; *Chu et al., 2014*).

Final luciferase values were normalized to $OD_{600}$ measurements of cultures to account for cell density. We note, however, that [*BIG+*] cells did not have a general growth advantage over naïve cells in SC-Ura, and when comparing optical density measurements to those counting cells using a hemacytometer, we observed no perturbation in the relationship between cell number and optical density for [*BIG+*] cells.

For wild strains, we considered the possibility that curing could reverse multiple epigenetic elements affecting plasmid copy number, transcription, or other elements of gene expression apart from protein synthesis. Indeed, after normalizing *Renilla* or firefly luciferase values to cell density, some strains have several-fold differences after curing, although they were closely correlated irrespective of which firefly codon variant was compared. Therefore, as for data presented in *Figure 6E*, for *Figure 8B* we also normalized firefly luciferase values to *Renilla* luciferase, which is expressed from the same plasmid. This normalization procedure thus tests for differences in protein synthesis that are codon-frequency dependent, that is, a measure of translational efficiency.

## Polysome profiling

Single colonies from two biological replicates per sample were used to inoculate 5 mL YPD cultures that were grown on a roller drum wheel at 30 °C overnight. Saturated cultures were added to 95 mL of YPD in 500 mL flasks and shaken at 225 rpm at 30 °C until cultures reached $OD_{600}$ 1.0. Five minutes prior to harvesting cells, we added cycloheximide (Sigma) to a final concentration of 100 µg/mL to arrest translation, by adding 1 mL of a 10 mg/mL stock solution (in ethanol) per culture, then immediately swirling flask and putting back on a shaker for 5 min 225 rpm 30 °C to permit the chemical to enter cells and arrest protein synthesis. Cultures were pelleted in 50 mL conical tubes for 3 min at 5000 rpm. After decanting the supernatant, pellets were quickly resuspended in ice-cold Polysome Lysis Buffer (PLB; *Jan et al., 2014*) (20 mM Tris pH 8.0, 140 mM KCl, 1.5 mM $MgCl_2$, 100 µg/mL cycloheximide, 1 % Triton X-100, RNase-free reagents), 250 µL total PLB per sample. Resuspended pellets were then flash-frozen in liquid nitrogen. Pellets were weighed to ensure their weights were near equal and then thawed on ice. 250 µL of additional ice-cold PLB was added per sample, making slightly over 0.5 mL per sample. Samples were then flash-frozen in tiny pellets ('yeast dippin' dots') by pipetting directly into a small dewar filled with liquid nitrogen and a wire mesh basket nested inside. Tiny pellets were then stored at –80 °C until lysis. Samples were lysed using a Retsch Cryomill (Haan, Germany) with 25 mL canisters and the following program: pre-cool, then 12 cycles of 15 Hz × 3 min. Smears of lysate were stained with Trypan blue and imaged under a microscope to verify efficient lysis. (We suspect with larger sample volume:canister volume ratios, fewer cycles would be necessary.)

Lysates were loaded onto 10–50% sucrose gradients pre-poured on a BioComp Gradient Master 108 (Fredericton, ND, Canada). Lysates generally contained RNA concentrations around 12–18 µg/µL. 30 µL of lysate was carefully pipetted onto the top of the sucrose gradient, and samples were spun in a Beckman SW41 Ti Rotor for 2.5 hr at 4 °C at 40,000 rpm. Gradients were analyzed on a Brandel fractionator (Gaithersburg, MD). Technical replicates (same lysate independently loaded onto separate gradients) showed a very high degree of similarity, as did biological replicates.

## [35S]-Methionine pulse labeling

Adapted from *Esposito and Kinzy, 2014*. Single colonies were inoculated into 5 mL complete media lacking methionine and grown overnight at 30 °C. The saturated starter culture was then inoculated into 75 mL of complete media lacking methionine to an $OD_{600}$ of 0.1 in a 250 mL flask. Flasks were grown at 30 °C shaking at 225 rpm until an $OD_{600}$ of 0.5–0.7 was reached at which point 37.5 µL of 100 mM (cold) methionine (Sigma) and 7.32 µL of 10.25 mCi/mL [35S]-methionine (PerkinElmer) was

added for a final concentration of 50 µM methionine and 1 µCi/mL [$^{35}$S]-methionine. At 15 min intervals from the time of methionine addition, OD$_{600}$ was determined by cuvette and 1 mL aliquots were obtained for scintillation counting. To the 1 mL aliquot, 200 µL 50 % TCA (Sigma) was added and incubated on ice for 10 min. The samples were then incubated at 70 °C for 20 min and briefly spun down. In a 1225 Sampling Manifold (MilliporeSigma XX2702550), 25 mm glass microfiber filters (GE Healthcare Life Sciences) were arranged and pre-wetted with 5% TCA. Samples were then applied to the glass fiber filters under vacuum. Filters were then washed twice with 5 mL 5% TCA and once with 95% ethanol. Filters were then removed from the manifold, air-dried on Whatman paper, placed in scintillation vials with 5 mL ScintiSafe Econo 1 scintillation cocktail solution (Fisher Scientific SX20-5), and underwent six rounds of technical replicated counting on a Beckman LS 600SCS. Scintillation counts were then normalized to OD$_{600}$.

## 2-D gel electrophoresis

Following 1 hr of pulse labeling with [$^{35}$S]-methionine, 10 mL of cells were collected and centrifuged at 16 k × g at 4 °C for 10 min. The supernatant was removed and 300 µL of TCA buffer was added (10 mM Tris-HCl pH 8.0; 10 % TCA; 25 mM NH$_4$OAc; 1 mM Na$_2$EDTA) along with glass beads. Samples were then vortexed for 1 min bursts with 5 min rests for five times at 4 °C. The slurry was transferred away from the glass beads to a fresh tube. 200 µL of fresh TCA buffer was added to beads to pick up residue. Samples were then split, one half destined for resuspension in SDS for protein quantification and the other in CHAPS for the 2-D gel. Samples were centrifuged at 16 k × g at 4 °C for 10 min. The supernatant was then removed from the resulting pellets and washed twice with 1 mL acetone. The samples were then resuspended in SDS Resuspension Solution (0.1 M Tris-HCl pH 11.0; 3% SDS) or CHAPS Resuspension Buffer (8 M urea; 4% CHAPS; 2% IPG Buffer 3–10; 40 mM DTT). CHAPS samples were then adjusted to 1 µg/µL. First-dimension resolution was conducted with 250 µg with 13 cm 3–10 NL Immobiline DryStrip gels (Cytiva). Rehydration protocol started with 12 hr rehydration followed by 500 V for 0.5 kVh, 1000 V for 1.0 kVh, 16,000 V for 14.5 kVh. First dimension strips were then trimmed to fit the second-dimension gel and equilibrated in Equilibration Buffer (6 M urea; 30 % glycerol; 2 % SDS, 50 mM Tris-HCl pH 8.8; 0.002 % bromophenol blue; 65 mM DTT) for 30 min. Strips were rinsed with MOPS running buffer and loaded onto 11 cm 4–12% Bis-Tris Criterion XT gels (Bio-Rad). Strips were sealed in place with an agarose plug (0.5 % agarose; 0.0002 % bromophenol blue) and ran at 200 V for 1 hr. Second-dimension gels were then stained in Coomassie and dried in a cellulose sandwich consisting of a plastic wrap on one side of the gel to facilitate the removal of one of the cellulose layers for autoradiography. X-ray film was then exposed for 9 days. Autoradiography spots were quantified in ImageJ by tracing out the spot and quantifying the integrated density of the signal. Densities of prion cells were then normalized to the naïve sample, as reported in *Figure 6I*.

## GFP-fusion measurements

Naïve or [*BIG$^+$*] cells were mated to the SWAT seamless-GFP library (*Weill et al., 2018*; *Yofe et al., 2016*) on solid YPD agar plates in 384-spot format for 24 hr at room temperature. Diploids were selected on media lacking both lysine and methionine (SD-Lys-Met) and propagated for 48 hr at room temperature. Diploids were inoculated into 60 µL of liquid media lacking both lysine and methionine (SD-Lys-Met) in 384-well plates. All library manipulations were carried out using a Singer ROTOR HDA robotic pinning instrument. Cells were propagated in liquid medium for 24 hr at 30 °C (OD$_{600}$ ~ 1), at which time OD$_{600}$ and green fluorescence were measured using a BioTek Synergy H1 plate reader. OD$_{600}$ was adjusted based on known blank wells, and the GFP/OD$_{600}$ measurements were normalized by Z-score ([$x_i$ − µ]/σ) within the naïve and [*BIG$^+$*] populations independently.

## Settings used for PWScan for predicted Pus4 target sites in silico

Matrix format: PFM-like matrix
Motif length: 17
Minimal log score: –10,000
Log scaling factor: 100
Genome assembly: sacCer3
p-value threshold: 1.000e$^{-05}$
Matrix score: –8265

Cut-off percentage: 93.88%
Background base composition: 0.31, 0.19, 0.19, 0.31
Search strand: both
Offset: 0
Non-overlapping matches: on

## Western blots

For SDS-PAGE, immunoblots, and protein yield measurements, cells were lysed using a Retsch Cryo-mill using the following program: six 3 min cycles at 15 Hz with 2 min cooling cycles in between. Cell lysates were loaded onto GenScript ExpressPlus SDS-PAGE 4–20% gels (Piscataway, NJ) and stained using Coomassie blue. For western blots, anti-FLAG M2 monoclonal antibody (Sigma) was used to detect 3XFLAG-Pus4, and anti-PGK1 monoclonal antibody (Invitrogen) was used to detect the loading control.

## RNA sequencing

Four independent colonies of naïve and [*BIG*$^+$] cells were grown in 5 mL liquid YPD cultures on a culture roller drum wheel overnight at 30 °C. Saturated cultures were diluted 1:1000 in new 5 mL YPD cultures and grown until OD$_{600}$ 1.0 (exponentially growing). Cultures were pelleted in a table-top centrifuge with the swing-bucket rotor at 4300 × g at 4 °C for 5 min, washing once with ice-cold 1 X PBS. RNA isolation, poly-A selection, the addition of ERCC spike-in controls, and sequencing were performed by the Beijing Genomics Institute. Analysis was performed using Kallisto (**Bray et al., 2016**) and Sleuth (**Pimentel et al., 2017**). Transcripts Benjamini–Hochberg-corrected *q*-value <0.05 and |log$_2$ fold-change| > 1 were considered differentially expressed. Raw data and other experimental information are available on the Gene Expression Omnibus (**Barrett et al., 2013**), accession: https://www.ncbi.nlm.nih.gov/geo/query/acc.cgi?acc=GSE176577.

## tRNA/rRNA quantification

Strains were grown up in 5 mL of YPD overnight at 30 °C. Saturated cultures were then diluted 15 × into fresh media and grown to an OD$_{600}$ 0.8. Cells were then pelleted, and total RNA was extracted by phenol-chloroform extraction. RNA concentration for each sample was then analyzed by Nanodrop and diluted to equal concentrations in RNase-free water. The extracted RNA was then run on a nucleic acid fragment analyzer, and total tRNA abundance was quantified relative to the abundance of 5.8 S rRNA.

### Pseudouridine measurements: high-resolution melt curve analysis

Protocol adapted from **Lei and Yi, 2017**. For each sample, 40 µg of total RNA was fragmented in RNA fragmentation buffer (New England Bio Labs, Ipswich, MA) at 94 °C for 3 min. Following ethanol precipitation, fragmented RNA was resuspended in 80 µL of 5 mM EDTA, denatured at 80 °C for 5 min, and then immediately chilled on ice. Each sample was then split into two 40 µL samples for a CMC-labeled and non-labeled control. The 40 µL RNA sample destined for CMC-labeling was added to 400 µL BEU + CMC buffer (50 mM bicine, pH 8.5; 4 mM EDTA; 7 M urea; 200 mM CMC [Sigma]). The non-labeled 40 µL RNA sample was added to 400 µL BEu buffer (50 mM bicine, pH 8.5; 4 mM EDTA; 7 M urea). Both samples were incubated at 37 °C for 20 min to carry out the CMC-Ψ reaction, followed by ethanol precipitation. Each sample was then resuspended in 200 µL Na$_2$CO$_3$ buffer (50 mM Na$_2$CO$_3$, pH 10.4; 2 mM EDTA) and incubated at 37 °C for 6 hr. Following incubation, RNA was ethanol precipitated and resuspended in 40 µL H$_2$O. RNA was annealed to primers by the addition of 4 µL 100 µM Random Hexamer Primers (TaKaRa, Mountain View, CA) and incubation at 65 °C for 5 min. Samples were chilled on ice afterward. To perform the reverse transcription, 32 µL RT Buffer (125 mM Tris, pH 8.0; 15 mM MnCl$_2$; 187.5 mM KCl; 1.25 mM dNTPs; 25 mM DTT) was added to each sample. Samples were then incubated at 25 °C for 2 min. After, 0.5 µL SuperScript II reverse transcriptase (SSII, Invitrogen, Waltham, MA) was added to each sample followed by incubation at 25 °C for 10 min, 42 °C for 3 hr, and 70 °C for 15 min. To perform qPCR analysis, 2 µL of the sample was mixed with 10 µL 2 X SYBR Mix (Kapa Biosystems, Wilmington, MA), 0.4 µL of each 10 µM primer (IDT, Coralville, IA), and 7.2 µL H$_2$O for a total of 20 µL for each reaction. qPCR was performed in a Bio-Rad CFX Connect Real-Time System (Bio-Rad, Hercules, CA) using the following protocol: initial

incubation at 95 °C for 5 min, followed by 45 cycles of 95 °C for 0.5 min and 60 °C for 1 min. Following amplification, the reaction was brought down to 54 °C and held for 5 s, increasing in temperature by 0.1 °C increments until 95 °C is reached to obtain melt curve data.

## Pseudouridine measurements: Sanger sequencing
Templates were *TEF1/TEF2* amplicons from the qPCR step above. Due to the short length of tRNA amplicons, the qPCR product was re-amplified with primers containing the restriction sites NcoI and HindIII and ligated into a pETm11 vector to allow for sequencing from the T7 promoter.

## Pseudouridine measurements: CLAP
Adapted from *Zhang et al., 2019*.

### CMC treatment of RNA
20 µg of total RNA in 12 µL of water was mixed with 24 µL of 1× TEU buffer (50 mM Tris-HCl pH 8.0, 4 mM EDTA, 7 M urea). Next, 1 M CMC was prepared fresh in TEU buffer, and 4 µL was added to +CMC samples while 4 µL of TEU buffer was added to –CMC samples for a total volume of 40 µL. To incorporate CMC, the samples were incubated at 30 °C for 16 hr. Following incubation, excess CMC was removed by the addition of 160 µL of 50 mM KOAc pH 5.5, 200 mM KCl, 3 µL 5 µg/µL GlycoBlue (Thermo Fisher, AM9515), and 550 µL of EtOH. The samples were incubated for 2 hr at –80 °C then centrifuged at 13 krpm at 4 °C for 35 min. The supernatant was removed and 500 µL of 75 % EtOH was added. The samples were incubated at –80 °C for 2 hr and centrifuged in the same manner as before. The EtOH precipitation and wash steps were repeated an additional time. After the second wash step, the samples were left at –80 °C overnight. The following day the samples were centrifuged again at 13 krpm at 4 °C for 35 min, and the supernatant was removed. To reverse the CMC-U/CMC-G adducts, the RNA pellet was mixed with 40 µL of 50 mM $Na_2CO_3$, 2 mM EDTA, and incubated at 37 °C for 6 hr. An additional 160 µL of 50 mM KOAc (pH 5.5), 200 mM KCl was added after incubation followed by ethanol precipitation.

### RNA 5′ phosphorylation and RNA-5 blocking oligo ligation
Performed similarly to *Zhang et al., 2019*. To reduce the signal derived from random RNA fragmentation during the CMC reaction, ±CMC-treated RNA in 6.5 µL $H_2O$ was mixed with 0.5 µL RNase inhibitor (NEB, M0307L), 1 µL of 10× T4 PNK reaction buffer, 1 µL of 1 mM ATP, 1 µL of T4 polynucleotide kinase (PNK), and then incubated at 37 °C for 30 min. The RNA-5 oligo (/5AmMC6/rArCrCrCrA; Integrated DNA Technologies) was ligated by addition of 1 µL of 10× T4 RNA Ligase Reaction Buffer, 1 µL of 100 µM RNA-5 oligo, 1 µL of 1 mM ATP, 1 µL of RNase inhibitor, 3 µL of DMSO, 2 µL of $H_2O$, and 1 µL of T4 RNA ligase I (NEB, M0437M), and incubated at 16 °C for 16 hr. The reaction was terminated by addition of 1.2 µL of 200 mM EDTA.

### Reverse transcription and splint ligation
Performed similarly to *Zhang et al., 2019*. Reverse transcription was carried out using 3 µL of the above ligation mixture (~1 µg RNA) using AMV reverse transcriptase (NEB, M0277L) and target-specific primers. To anneal the primer, 1 µL of 10× annealing buffer (250 mM Tris-HCl [pH 7.4], 480 mM KCl) and 1 µL of 0.5 µM target-specific RT primer were added, and the mixture was incubated at 93 °C for 2 min. The RT reaction was performed by mixing 5 µL of 2× AMV RT reaction mixture to make 0.6 U/µL AMV RT, 1× AMV RT buffer, and 0.5 mM of each dNTP. The RT reaction was run at 42 °C for 1 hr. RT enzyme was inactivated by incubation at 85 °C for 5 min, and then 1 µL of 5 U/µL RNase H (NEB, M0297L) was added and the mixture incubated at 37 °C for 20 min to digest the RNA. RNase H was inactivated by incubating the mixture at 85 °C for 5 min.

To anneal the adaptor, 1 µL of the adaptor/splint oligos mixture (1.5 µM each) was added to the above RT mixture, and then incubated at 75 °C for 3 min. To ligate the adaptor, 4 µL of 4× ligation mixture was added for a final concentration of 10 U/µL of T4 DNA ligase (NEB, M0202L), 1× T4 DNA ligase reaction buffer, and 12.5 % DMSO. The ligation was performed at 16 °C for 16 hr. DNA ligase was inactivated by incubating the mixture at 65 °C for 10 min.

## PCR amplification and gel electrophoresis

To perform PCR, 2 µL of the above ligation mixture was mixed with various components for a final condition of 1× Q5 reaction buffer, 1× Q5 high GC enhancer, 200 µM of each dNTP, 0.5 µM forward and reverse primers, and 0.02 U/µL Q5 high-fidelity DNA polymerase (NEB, M0491L); the final PCR reaction volume was 35 µL. PCR was done for 35 cycles. A PCR cleanup was performed, and 22 µL of water was used to elute the final product.

About half of the PCR product, 9 µL, was mixed with 7 µL of water and 4 µL of 6× TriTrack DNA Loading Dye (Thermo Fisher, R1161). The entire mixture was loaded on a prerun 10 % nondenaturing gel containing 1× TBE, together with Ultra Low Range DNA ladder (NEB, N0558S). The gel was stained with SYBR gold nucleic acid gel stain (Thermo Fisher, S11494) for 10 min. Product bands were visualized using the Amersham Typhoon imaging system and the bands quantified using ImageQuant TL.

## Data display

Plots/graphs were made using PRISM 7/8/9 software (GraphPad, San Diego, CA).

## Acknowledgements

We gratefully acknowledge Rebecca Freilich and Alan Itakura (Stanford University), Mike Harms (University of Oregon), and Gabriel Neurohr (ETH Zürich) for their review of the manuscript and members of the Jarosz laboratory for helpful discussions. We thank Maya Schuldiner (Weizmann Institute) for the generous gift of the SWAT GFP library; Kathrin Leppek, Gerald Tiu, and Maria Barna (Stanford University) for assistance with polysome gradients and profiling and the use of their instrument; Huili Guo (IMCB Singapore) for advice on presentation of polysome profiling data; José Aguilar Rodriguez (Stanford University) for creating the Position Weight Matrix and scanning the yeast genome for candidate pseudouridylation sites; and the Genomics and Cell Characterization Core Facility (University of Oregon) for assistance with tRNA quantification.

## Additional information

### Competing interests

Matt Kaeberlein: Reviewing editor, *eLife*. The other authors declare that no competing interests exist.

### Funding

| Funder | Grant reference number | Author |
| --- | --- | --- |
| National Institute of General Medical Sciences | F32-GM109680 | David M Garcia |
| Burroughs Wellcome Fund | Postdoctoral Enrichment Program 1015119 | David M Garcia |
| National Institute on Aging | P30AG013280 | Matt Kaeberlein |
| National Science Foundation | Graduate Research Fellowship | Edgar A Campbell |
| National Institute of General Medical Sciences | F32-GM125162 | Christopher M Jakobson |
| National Institutes of Health | DP2-GM119140 | Daniel F Jarosz |
| National Science Foundation | NSF-CAREER-MCB116762 | Daniel F Jarosz |
| Kinship Foundation | Searle Scholar Award (14-SSP-210) | Daniel F Jarosz |
| Sidney Kimmel Foundation | Kimmel Scholar Award (SFK-15-154) | Daniel F Jarosz |

| Funder | Grant reference number | Author |
|---|---|---|
| David & Lucile Packard Foundation | Science and Engineering Fellowship | Daniel F Jarosz |
| Ford Foundation | Postdoctoral Fellowship | David M Garcia |
| Donald E. and Delia B. Baxter Foundation | Award | David M Garcia |

The funders had no role in study design, data collection and interpretation, or the decision to submit the work for publication.

## Author contributions

David M Garcia, Conceptualization, Data curation, Formal analysis, Funding acquisition, Investigation, Methodology, Project administration, Resources, Supervision, Validation, Visualization, Writing – original draft, Writing – review and editing; Edgar A Campbell, Christopher M Jakobson, Conceptualization, Data curation, Formal analysis, Funding acquisition, Investigation, Methodology, Resources, Software, Validation, Visualization, Writing – original draft, Writing – review and editing; Mitsuhiro Tsuchiya, Data curation, Formal analysis, Investigation, Methodology, Resources, Validation, Visualization; Ethan A Shaw, Investigation, Methodology, Resources, Validation, Visualization; Acadia L DiNardo, Investigation, Resources; Matt Kaeberlein, Funding acquisition, Supervision, Writing – review and editing; Daniel F Jarosz, Conceptualization, Data curation, Formal analysis, Funding acquisition, Investigation, Methodology, Project administration, Resources, Software, Supervision, Validation, Visualization, Writing – original draft, Writing – review and editing

## Author ORCIDs

David M Garcia http://orcid.org/0000-0003-0600-9527
Christopher M Jakobson http://orcid.org/0000-0001-7594-7416
Acadia L DiNardo http://orcid.org/0000-0003-3219-4105
Matt Kaeberlein http://orcid.org/0000-0002-1311-3421
Daniel F Jarosz http://orcid.org/0000-0003-3497-5888

## Decision letter and Author response

Decision letter https://doi.org/10.7554/eLife.60917.sa1
Author response https://doi.org/10.7554/eLife.60917.sa2

# Additional files

## Supplementary files

• Supplementary file 1. See .docx file for parameter values used for the competitive fitness models shown in *Figure 2* and *Figure 2—figure supplement 1*.

• Supplementary file 2. RNA-sequencing results. See .xlsx file for transcripts per kilobase million for each gene in four replicates of naïve and four replicates of [*BIG*$^+$]. Genes that showed significant differences are listed in a separate tab.

• Supplementary file 3. Proteins whose expression is changed in [*BIG*$^+$] cells.

• Supplementary file 4. Yeast strains, plasmids, and oligonucleotides used in this study. See .xlsx file.

• Supplementary file 5. Yeast genes with predicted Pus4 target sites based on scanning the yeast genome using a position weight matrix. See .csv file.

• Transparent reporting form

• Reporting standard 1. Statistical reporting.

## Data availability

Updated sequencing data have been deposited in GEO under accession code GSE176577.

The following dataset was generated:

| Author(s) | Year | Dataset title | Dataset URL | Database and Identifier |
|-----------|------|---------------|-------------|-------------------------|
| Garcia DM, Campbell EA, Jakobson CM, Tsuchiya M, Shaw EA, DiNardo A, Kaeberlein M, Jarosz DF | 2021 | A prion accelerates proliferation at the expense of lifespan | https://www.ncbi.nlm.nih.gov/geo/query/acc.cgi?acc=GSE176577 | NCBI Gene Expression Omnibus, GSE176577 |

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

## Appendix 1

We also examined the relationship between the proteins whose expression was altered in [*BIG*⁺] cells and other features related to gene expression, including mRNA secondary structure and pseudouridylation. Comparing the altered gene set to yeast transcripts with more double-stranded regions (*Kertesz et al., 2010*), no relationship emerged. We also compared the gene list to mRNAs that have been experimentally validated to be pseudouridylated by Pus4 (*Carlile et al., 2014*; *Lovejoy et al., 2014*; *Schwartz et al., 2014*), but did not see any significant correlation or changes among the limited number targets that contained Pus4-dependent pseudouridylation sites in these studies, and which were present in the SWAT gene collection (13 genes overlapped these two categories). Given the limited reproducibility of mapping pseudouridylation sites transcriptome-wide in yeast from past studies, with large variability sample to sample (*Safra et al., 2017*; *Zaringhalam and Papavasiliou, 2016*), it is possible that the earlier studies missed possible Pus4 mRNA targets that could have been present in the list of altered genes. Therefore, we also created a position weight matrix (PWM) from an alignment of sequences centered around pseudouridine sites in tRNAs and mRNAs that were experimentally verified in the previous studies (41 sequences of 17 nucleotides in length), and then used this PWM to scan the yeast genome for other predicted Pus4 target sites. The scan produced 433 hits (*Supplementary file 5*), which, when compared to 127 proteins that changed from the SWAT gene collection, overlapped by only five genes. Therefore, we conclude that the changes we observe for the translation of ~130 messages cannot be explained by Pus4-dependent pseudouridylation of their mRNAs.

We also did not observe any relationship between genes with increased or decreased protein levels in [*BIG*⁺] and isoelectric point (pI), protein length, protein half-life, or GO category enrichments (process, component, or function). We did however observe several properties that were altered in both the genes that went up and those that went down: hydropathy score (GRAVY) (*Kyte and Doolittle, 1982*), aromaticity, and CAI, codon bias, and frequency of optimal codons. We observed some amino acid enrichments as well, with genes going up in [*BIG*⁺] being depleted in leucine, phenylalanine, and proline, and those down depleted in phenylalanine and aspartic acid.

Finally, given that many of the effects that we measured were post-transcriptional, we also examined genes with altered GFP levels (significantly up or down) in [*BIG*⁺] cells using the MEME Suite, specifically for the enrichment of predicted binding sites for RNA binding proteins (*Bailey et al., 2009*). We found that binding sites for the CCR4-NOT deadenylase complex were significantly enriched in our hits (motif alt ID CNOT4; consensus motif ACACAWA; adjusted p-value 0.0135).

