## [Decision Letter]

**Acceptance summary:**

This manuscript reports that pseudouridine synthase Pus4 acts as a prion switch that regulate the epigenetic state of cell growth by promoting protein synthesis. This is a very interesting study backed by a plethora of supporting data.

**Decision letter after peer review:**

Thank you for submitting your article "A prion accelerates proliferation at the expense of lifespan" for consideration by *eLife*. Your article has been reviewed by 3 peer reviewers, one of whom is a member of our Board of Reviewing Editors, and the evaluation has been overseen by Jessica Tyler as the Senior Editor. The reviewers have opted to remain anonymous.

The reviewers have discussed the reviews with one another and the Reviewing Editor has drafted this decision to help you prepare a revised submission.

As the editors have judged that your manuscript is of interest, but as described below that additional experiments are required before it is published, we would like to draw your attention to changes in our revision policy that we have made in response to COVID-19 (https://elifesciences.org/articles/57162). First, because many researchers have temporarily lost access to the labs, we will give authors as much time as they need to submit revised manuscripts. We are also offering, if you choose, to post the manuscript to bioRxiv (if it is not already there) along with this decision letter and a formal designation that the manuscript is "in revision at eLife". Please let us know if you would like to pursue this option. (If your work is more suitable for medRxiv, you will need to post the preprint yourself, as the mechanisms for us to do so are still in development.)

Summary:

This manuscript reports that pseudouridine synthase Pus4 acts as a prion switch that regulate the epigenetic state of cell growth by promoting protein synthesis. The authors demonstrated that transiently overexpression of Pus4 induced an epigenetically controlled prion-like state that showed faster growth but shorter lifespan. These phenotype changes appeared to be linked to altered protein synthesis, a mechanistic insight that explains both the faster growth and shorter lifespan phenotypes. All reviewers are impressed with the study and think that this manuscript is suitable for *eLife*. However, they also raised several concerns that should be addressed before this manuscript is accepted for publication in *eLife*.

Essential revisions:

1) The abstract and introduction lack deepness and incorporate some unclear concepts. For example:

Line 16 – The opening sentences of the abstract introduce a jargon that is not true for organismal aging. That can be true in the single cell level for some cell types and for some organisms, however; there is no general concept in the aging field that organisms are evolved to select one of these strategies. This sentence was not clarified in the introduction part with further details on the origin of this statement along with necessary references.

Line 18: "Lifespan is often inversely correlated with cell size and proliferation". Similarly, this argument in the abstract was not explained in the introduction. Although, there are references regarding to the inverse correlation of cell size and lifespan, there has not been much argument/discussion and related reference about inverse correlation of proliferation rate and lifespan. In fact, in yeast, there is no defined general concept that describes inverse correlation between proliferation rate and aging. More clarification is needed on this statement.

2) All three reviewers pointed out some issues with Figure 4. Please address reviewers' concerns there.

3) One reviewer had some concern about the data presented in Figure 5B. Please address this issue in your revision.

4) Reviewer 1 raised some issues with the RNA-seq analysis. Please address his/her questions on.

5) Two reviewers have some concerns in Figure 6. Please address these concerns.

6) Reviewer 2 pointed out the weakness in the analysis presented in Figure 8. Please consider addressing this concern.

7) Reviewer 2 also pointed out some issues with discussion, which should be considered to improve the clarify of the paper.

*Reviewer #1:*

In the manuscript titled "A prion accelerates proliferation at 1 the expense of lifespan", authors Garcia et al., reported that pseudouridine synthase Pus4 acts as a prion switch that regulate the epigenetic state of cell growth by promoting protein synthesis. Overall, this is quite an intriguing paper to read. Most of the experiments were well thought out and designed, most of the data presented are clear and support their conclusions. However, there are a few issues that should be further addressed or clarified before it is accepted for publication.

1. In Figure 4G, it does appear that the cytoplasmic localization of Pus4 in [BIG+] cells is ER, maybe rough ER, which may support its function in mRNA pseudouridinylation and promoting protein synthesis. This may be further verified by using inducing [BIG+] in GFP-tagged ER marker or rough ER marker strains.

2. The data in Figure 5B are the least clear and convincing in this paper. It is difficult to tell the difference between [BIG+] and pus4del. These data should be quantified in the same way as Figure 7.

3. The assessment of RNA levels in [BIG+] cells is the biggest issue in this manuscript. RNA-seq from bulk cell sample should be used with spike-in controls normalizes to cell count in order to detect global mRNA level changes (PMID: 23101621). Also, it has been shown clearly that cell size is positively correlated with RNA content (PMID: 30718850). Hence, the levels of RNA or mRNA in each cell should be carefully determined, either by bulk RNA-seq with spike-in control, or single cell RNA-seq.

4. Figure 6 shows that [BIG+] cell growth was not efficiently inhibited by rapamycin and that [BIG+] cells have higher levels of polysomes. Hence, it should be tested whether TOR signaling is hyperactivated by [BIG+] and that whether this hyperactivation can be efficiently inhibited or attenuated by rapamycin. This can be done by analyzing the phosphorylation levels of Sch9, a target of TOR.

*Reviewer #2:*

The manuscript by Garcia et al., demonstrates a unique case of post-transcriptional modification "pseudouridylation", on growth adaptation and its effect on lifespan. This modification is induced by transient overexpression of the highly conserved pseudouridine synthase PUS4/TRUB. Authors demonstrated that prion like state of the PUS4 gene affects yeast cell physiology and phenotypes including, growth rate, cell size and lifespan. Authors also connected these phenotype changes to translational regulation of small number of proteins with rare codons. Interestingly these phenotypes are altered without changing mRNA level, including total tRNA and ribosomal RNAs. In addition, global level of protein translation and/or abundance was not affected by this process. They concluded these observations in the light of several carefully designed and elegant experiments. Overall, findings are interesting, and manuscript is written well. As such, the manuscript will greatly promote progress in the field of studying the molecular and cellular mechanisms of aging. However, there are several points to be clarified/improved, which will further strengthen the study. Therefore, the manuscript is suitable for *eLife*, and I recommend its publication after the following revisions are addressed.

1 – The abstract and introduction lack deepness and incorporate some unclear concepts. For example:

Line 16 – The opening sentences of the abstract introduce a jargon that is not true for organismal aging. That can be true in the single cell level for some cell types and for some organisms, however; there is no general concept in the aging field that organisms are evolved to select one of these strategies. This sentence was not clarified in the introduction part with further details on the origin of this statement along with necessary references.

Line 18: "Lifespan is often inversely correlated with cell size and proliferation". Similarly, this argument in the abstract was not explained in the introduction. Although, there are references regarding to the inverse correlation of cell size and lifespan, there has not been much argument/discussion and related reference about inverse correlation of proliferation rate and lifespan. In fact, in yeast, there is no defined general concept that describes inverse correlation between proliferation rate and aging. More clarification is needed on this statement.

2 – Line 727: Authors claim "These observations suggest that epigenetic, and potentially prion-mediated control of mechanisms like the [BIG+] prion that we have described here, may be widespread in nature". Out of 23 wild yeast isolates, authors only tested four of them and showed the cell size differences. Out of these 4 isolates, only 2 of them showed decreased translation. Authors are clearly cherry picking the results to convince that the observed BIG phenotype is widespread. From the data they present, it is not true. These wild isolates have been previously characterized to have many SNPs and different phenotypes under different stress conditions. Transient inhibition of Hsp70 may cause differences in cell phenotypes, including cell size by acting on different mechanism. The direct testing of BIG phenotype should be performed through transient expression of PUS4 in these strains and measuring the cell size. Also, strains with increased cell size should be subjected to RLS or CLS measurement to align with authors' conclusion. In addition, YPD medium specificity of BIG phenotype already indicates that it is not widespread and it is condition specific. I do not think any of these wild isolates' niches were similar to the YPD condition in their natural environment. Overall, this part is very weak and with the current version, it does not add any further strength to the findings.

3 – Authors conclude that this newly identified prion-based mechanism engages an altered translational program to favor a 'live fast, die young' strategy. There are several interesting findings in this manuscript, however they are somehow disconnected from each other. Basically how these findings are complementing to each other and to their role in observed phenotype need deeper discussion. In the discussion part, authors are only focusing on altered modification of TEF1/TEF2 elongation factors. Some of their findings have not been mentioned. For example, findings of 130 differentially expressed proteins with decreased codon adaptation index along with the findings of no decrease in global protein level are very interesting results. However, the authors seem to avoid discussing these findings. In addition, their findings about [BIG+] cells enhancing translation of some proteins, especially those containing a greater frequency of rare codons need more highlight and to be connected to other findings.

*Reviewer #3 :*

The authors report the discovery and characterization of [BIG+] a new yeast prion which shares some properties with other well-studied yeast prions, but also shows some differences, for example in chaperone dependency and physical state. That this prion can apparently control cell growth behaviour of both lab and wild strains of the organism is intriguing, but the precise mechanism by which this is achieved is not fully established in this study. In ruling out any impact on transcription and steady state mRNA/tRNA levels the focus is on a potential post-transcriptional mechanism involving differential synthesis of a relatively large number of proteins. Among the proteins identified there are a number of candidates that could explain the observed [BIG+] phenotypes of cell size and life span changes, but we are left with a list without any one in particular being pinpointed or directly linked to the [BIG+] phenotype.

There are two issues where I have some concerns:

1. The authors provide several lines of evidence that [BIG+] impacts on the translation of a subset of mRNAs, but I felt the analysis (cf Figure 6) was of relatively low resolution or indirect. Using a coomassie stain of a 1D SDS-PAGE (Fig6G) analysis is a very low resolution approach to look for global differences in protein levels and it would be relatively straightforward – and more informative – if a high res analysis was undertaken. For example, a 2D-SDS PAGE study using pulse-35S-Met-labelled proteins would identify any major changes in the rates of synthesis of at least the higher abundance proteins in [BIG+] vs [big-] cells and give strong evidence for the proposed altered translational programme. The approach they do use – although a clever one – uses quantitation of GFP fluorescence using an N terminally located GFP for 4000+ different proteins. Ideally one would have liked to have seen some evidence that those proteins identified by this screen also showed differences in levels in their native form i.e. without GFP fusions. Also, the shift in ribosome loading shown in Figure 6H; could the increase in the size of polysomes reflect a defect in translation termination?

2. From the data shown it is not clear what the physical form of the Pus4 protein is, in [BIG+] cells. In a relatively low resolution image of Pus4-GFP fusions (Figure 4G) the authors argue that there is a "substantial fluorescence in a fragmented network throughout the cytoplasm" only in [BIG+] cells but to make this statement we need to see higher res images. I was not convinced by the images shown. If they do exist they are clearly not the only form seen in [BIG+] cells as in Figure 8C a wild strain shows Pus4-GFP puncta. Why this difference? Furthermore, the majority of Pus4 in the [BIG+] lab strain is located in the nucleolus; how do we know this is the nucleolus and what happens to the contents of the nucleolus in the cytoduction experiments reported here? A more biochemical approach into analysis of the state of the Pus4 protein is also needed using the methods applied to the study of other yeast prions?

---

## [Author Response]

Essential revisions:1) The abstract and introduction lack deepness and incorporate some unclear concepts. For example:Line 16-The opening sentences of the abstract introduce a jargon that is not true for organismal aging. That can be true in the single cell level for some cell types and for some organisms, however; there is no general concept in the aging field that organisms are evolved to select one of these strategies. This sentence was not clarified in the introduction part with further details on the origin of this statement along with necessary references.Line 18: "Lifespan is often inversely correlated with cell size and proliferation". Similarly, this argument in the abstract was not explained in the introduction. Although, there are references regarding to the inverse correlation of cell size and lifespan, there has not been much argument/discussion and related reference about inverse correlation of proliferation rate and lifespan. In fact, in yeast, there is no defined general concept that describes inverse correlation between proliferation rate and aging. More clarification is needed on this statement.

Thank you for these suggestions to improve the clarity and framing of the problem in our opening paragraphs. We have now revised the abstract and the beginning of the Introduction.

For example, in the previous version of the manuscript, the abstract began:

“Organisms often commit to one of two strategies: living fast and dying young or living slow and dying old. In fluctuating environments, however, switching between these two strategies could be advantageous. Lifespan is often inversely correlated with cell size and proliferation, which are both limited by protein synthesis.”

Has now been changed to:

“In fluctuating environments, switching between different growth strategies, such as those affecting cell size and proliferation, can be advantageous to an organism. Trade-offs arise, however. Mechanisms that aberrantly increase cell size or proliferation— such as mutations or chemicals that interfere with growth regulatory pathways—can also shorten lifespan. Here we report a natural example of how the interplay between growth and lifespan can be epigenetically controlled."

We have made other changes to the Introduction, which we hope will further increase the clarity of our statements and framing.

2) All three reviewers pointed out some issues with Figure 4. Please address reviewers' concerns there.

All three reviewers indicated that it was essential to improve the quality of the Pus4-GFP micrograph in Figure 4G, and we thank them for this suggestion. Reviewer #2 also added the specific comment above about the possible "overload" of Pus4 protein. We have revised the panel with a new image.

Some steps we took that improved the image quality include using an agar pad and employing background correction, the details of which are provided in the Materials and methods lines (1390–1395 and 1412–1418).

In response to reviewer #2’s specific comment about “overloading” of Pus4p, we want to clarify that Pus4 is not being overexpressed in this experiment: the protein is seamlessly tagged and subject to endogenous transcriptional regulation (Yofe et al., 2016, PMID: 26928762).

3) One reviewer had some concern about the data presented in Figure 5B. Please address this issue in your revision.

We thank the reviewer for the suggestion to quantify the tRNA pseudouridylation as we did for mRNA pseudouridylation. We have added revised data to Figures 5B–D, described on lines 317–332. This analysis shows that tRNAs remain pseudouridylated in [*BIG*^+^] cells.

To complement this analysis, we also performed an additional experiment, investigating pseudouridylation of U55 in tRNAs using a different method (now included as Figure 5—figure supplement 1A–B and discussed on lines 332–339). We sub-cloned the PCR amplicons from our +CMC and -CMC treated samples and analyzed them by Sanger sequencing. We observed no mutations in the TΨC loop in non-CMC treated/”unlabeled” samples. In CMC-treated samples from both naive and [BIG+] cells, we observed mutations surrounding the U55 site, characteristic of lesions generated by CMC adduct formation at Ψ55. We did not observe any such mutations in CMC-treated samples from cells lacking the Pus4 protein, confirming that the assay is specific to Pus4 activity. These data provide further evidence that a major substrate of Pus4, tRNA, is pseudouridylated in [*BIG*^+^] cells.

4) Reviewer 1 raised some issues with the RNA-seq analysis. Please address his/her questions on.

Reviewer #1 expressed concerns that our RNA-sequencing analysis may not have controlled sufficiently for systematic changes to mRNA expression levels between naïve and [*BIG*^+^] cells, as our measurements were from bulk cell populations. They suggested that adding spike-in controls would be one way to improve our ability to report quantitative changes.

To address this concern, we prepared new samples and analyzed them by RNA-sequencing with spike-in controls, permitting us to validate that there was quantitative scaling in gene expression within each sample. The ERCC spike-in standards exhibited a linear relationship between mass and estimated transcripts per million (TPM) across the entire range of TPM values included in our gene-level analysis. We identified 19 significantly upregulated (log2 fold-change > 1 and Benjamini-Hochberg corrected q value <0.05) genes and 28 significantly downregulated genes in [*BIG*^+^] cells. We have replaced the previous RNA-sequencing data with these data in the manuscript. We added Figure 5—figure supplement 1C to show the quantitative scaling of our spike-in controls.

5) Two reviewers have some concerns in Figure 6. Please address these concerns.

Our response can be broken up into two parts:

1. Reviewer #3 expressed concern about our conclusion that translation is altered in [*BIG*^+^], mainly since we had not performed experiments that directly measure the rate of protein synthesis.

2. Reviewer #1 asked us to measure phosphorylation levels on Sch9, to mechanistically bolster our observation that [*BIG*^+^] cells are more resistant to rapamycin, an inhibitor of Tor, an Sch9 kinase.

Measuring the rate of protein synthesis:

In response to Reviewer #3’s concern regarding how translation is affected, we have performed their suggested ^35^S-methionine labeling experiment and 2-D gel electrophoresis and thank them for the suggestion. These data are now described in lines 449–455 and 484–494. The rate of new total protein synthesis is indeed higher in [*BIG*^+^] cells than in naïve cells, consistent with other results in our manuscript. We have added these data as Figure 6G of the revised manuscript.

The 2-D gel analysis offered more evidence of an altered translation program. Our efforts to keep the ^35^S-methionine pulse brief, the low level of many proteins affected by [*BIG*^+^], and limitations on the total amount of protein that could be run on our gel apparatus restricted the number of proteins that could be observed. Nonetheless, we were able to reproducibly visualize thirteen radiolabeled proteins. Following a ^35^S-methionine pulse in [*BIG*^+^] cells the signal for seven of these proteins increased, one decreased, and five remained very similar (Figures 6H–I). These data provide further support for suggestion that [*BIG*^+^] cells have an altered translation program, with variations in even the most rapidly synthesized proteins. This was consistent with results we previously reported from the GFP screen.

Measuring the phosphorylation of Sch9/TOR activity:

We also addressed TOR activity by investigating phosphorylation of Sch9, a principal target of the TOR kinase (Urban et al., 2007 (PMID: 17560372)). We have conducted this assay in consultation with two labs that have used it extensively. The assay relies on chemical fragmentation of a tagged version of Sch9 protein, followed by resolving the fragments using PAGE and Western Blot analysis. Sch9 protein is chemically fragmented using NTCB, which cyanylates cysteine residues. Under alkaline conditions, this leads to cleavage at the modified residues and release of a ~50kD C-terminal fragment of Sch9, which both permits Western blot analysis because the HA-tag is on the C-terminus, and theoretically permits better separation of differentially phosphorylated forms using PAGE. Six Tor1-dependent phosphorylation sites are within this C-terminal fragment.

Despite many attempts at this assay, it has not worked well for us in our strains. We do not see the characteristic multiple fragments corresponding to differentially phosphorylated forms of Sch9, the patterns of which have been reported to change when cells are treated with TOR pathway effectors, rapamycin or cycloheximide (Urban et al., 2007).

In Figure 1A from Urban et al., 2007 they used the Sch9 cleavage assay to show that chemicals that perturb activity of the TOR pathway alter the pattern of Sch9 phosphorylation. They observed at least five distinct bands centered around the 50kD marker band.

Lane 5: untreated cells

Lane 6: treated with Rapamycin for 30m—this inhibits Tor1 which leads to rapid reduction in Sch9 phosphorylation, collapsing the series of differentially phosphorylated proteins to a completely unphosphorylated form that runs faster through the gel matrix.

Lane 7: inhibition of translation using cycloheximide has the opposite effect of rapamycin treatment—this triggers the cell to increase growth to compensate for reduced translation, and since more phosphorylation of Sch9 corresponds with increased protein synthesis, this leads to an increase in the ratio of fully-phosphorylated Sch9 compared to less phosphorylated forms, and thus more intense bands higher up in the gradient of bands centered around 50kD.

Lane 8: Wortmannin, a kinase inhibitor (thus Tor1 inhibitor), has a similar effect as rapamycin treatment: reduced phosphorylation of Sch9 and more intense lower bands.

Author response image 1 is a representative blot of NTCB-treated Sch9 protein from our many attempts to replicate this assay. We used the same concentrations of cycloheximide and rapamycin used in the Urban *et al.,* study. We have attempted many different gel running conditions and varied lysis conditions with input from other investigators knowledgeable about this technique. We have also modified our strains to make them prototrophic, as was done in the Urban *et al.,* study. We also note that in parallel with these experiments, we have successfully reproduced the effects of these drugs in inhibiting cell growth.

**Author response image 1. sa2fig1:** 

Despite this, even in our control naïve BY4741 strains, we do not see the multiple distinct bands observed in the earlier study in untreated cells, nor do we see an expected shift in the bands in rapamycin-treated cells. Thus, we cannot conclude from this assay whether the phosphorylation status of Sch9 is different in [*BIG*^+^]. Encouragingly, with cycloheximide treatment, we did observe a characteristic shift upward of the C-terminus, indicating increased phosphorylation as seen in the prior publication. However, it is not to the same extent as previously reported and because of the lack of distinct phosphorylated forms, it is challenging to evaluate whether there is a difference in Tor1 activity in [*BIG*^+^] using this assay. As an alternative, we also ran our lysates on “Phos-tag” gels, which contain chemicals that exacerbate differences in the migration of phosphorylated polypeptides (causing them to run more slowly than in standard PAGE gels). Unfortunately, these have also not resolved the relevant differentially phosphorylated forms.

To conclude, we have committed significant resources in this assay, and consulted extensively with experts in them, but have nonetheless been unable to reproduce the variety of differentially phosphorylated forms of Sch9. It is therefore challenging to conclude much from these experiments about potential differences in Tor1 activity in naïve and [*BIG*^+^] cells — our data neither support nor refute potential differences. Some of the discrepancies may arise from multiple differences in genetic background between our strain (BY/S288C) and those used in prior papers reporting this assay. However, chasing these down would take a significant amount of time and offers no guarantee of success. We therefore turned to alternative approaches to assess TOR activity.

Our alternative approach for analysis of TOR activity in [*BIG*^+^]:

We have used another method that is complementary to the Sch9 phosphorylation analysis and has proved informative about differences in the TOR pathway in [*BIG*^+^] cells. We examined a protein whose localization depends on Tor1 activity, Par32 (*p*hosphorylated *a*fter *r*apamycin 32). Par32 is a downstream effector that is phosphorylated by Npr1 in response to Npr1's phosphorylation by Tor1. Changes to Par32 localization can therefore be linked to changes in Tor1 activity. Normally present mostly in the plasma membrane, inhibition of the TOR pathway causes Par32 to move away from the plasma membrane (Varlakhanova et al., PMID: 30156471). To examine whether Par32 localization is altered in [*BIG*^+^] cells relative to naïve controls, we crossed these strains to a partner containing a seamlessly GFP-tagged *PAR32* gene. We then measured the fraction of Par32 protein localized to the plasma membrane and quantified this ratio for many cells. In naïve cells, we observed a fraction of membrane Par32 signal that matched the previous study (~1.3). In [*BIG*^+^] cells, we observed a significant increase in this ratio, an average 60% higher value. This result would be consistent with the higher TOR activity in [*BIG*^+^] cells, possibly explaining why they are more resistant to rapamycin (Figure 6B). We have added these new data to the revised manuscript, in Figure 6—figure supplement 1A.

For responses to other more specific comments regarding data in Figure 6, see below directly under reviewer prompts.

6) Reviewer 2 pointed out the weakness in the analysis presented in Figure 8. Please consider addressing this concern.

Reviewer #2 explained that they found our claim of [*BIG*^+^] being in wild yeast strains weakened by the small number of the 23 wild isolate strains tested with phenotypes consistent with the prion. Likewise, they pointed out that the way we determined these traits were epigenetic was via transient chaperone inhibition. Although consistent with a prion-based phenotype, this experiment does not rule out other prions that do not depend upon Pus4.

We have now performed additional experiments and modified our language in this section to address the reviewer’s concern. We intended to identify whether a [*BIG*^+^]-like state exists in wild strains. Indeed, because of the conditional benefit of [*BIG*^+^], and based on our prior work identifying prions in wild fungal isolates, we did not expect that most of them would harbor this element, or another like it. However, the California wine strain BC187 had some features that resemble those we observed for strains harboring [*BIG*^+^] that we studied in the laboratory. We now provide further experimental evidence supporting a [*BIG*^+^]-like mechanism in BC187 cells.

In the original submission, we showed that BC187 exhibited permanent cell size changes, changes in translation, and differences in Pus4 expression after transient inhibition of Hsp70. Reviewer 2 suggested that we subject chaperone ‘cured’ BC187 cells to transient Pus4 overexpression to reverse the ‘curing’ process and restore a [*BIG*^+^]-like state, an excellent suggestion that we followed. When we transiently overexpressed Pus4 in BC187 cells that had been previously ‘cured’ of [*BIG*^+^], the cells returned to their original “large” size (Figure 8D, revised manuscript). The same manipulation did not affect the already large BC187 lineages that had not been subject to transient Hsp70 inhibition (Figure 8—figure supplement 1, revised manuscript). Therefore, the specific overexpression of Pus4 alone can recreate a [*BIG*^+^]-like state in wild yeast, as it does in laboratory yeast.

Reviewer 2 also recommended that we measure RLS and CLS in BC187 before and after transient chaperone inhibition. We attempted a CLS experiment, growing cells for more than two months, but unfortunately, the cultures completely evaporated before we could detect a significant reduction in viability. From work by Kaya *et al.,* 2015 (PMID: 27030810), it is known that BC187 wild yeast strains are very long-lived compared to laboratory *S. cerevisiae*.

The RLS experiments have also not been possible to pursue because they are performed in close quarters by a team of students, and restrictions from the University of Washington (where our RLS experiments were performed) prohibit doing them at this time. Due to the extended timeline of these experiments and uncertainty about when institutional regulations might permit them, we unfortunately do not foresee completing them in a timeframe consistent with other essential revisions that were requested.

Our additional data nonetheless demonstrate that transient overexpression of Pus4 alone leads to changes in cell size in wild yeast strain BC187 that are curable by transient Hsp70 inhibition. We agree that finding more examples in other strains will be essential to claim that this is "widespread" and have softened our language accordingly. However, our success in finding the element after screening just twenty strains suggests it may be as frequent, if not more so, as multiple other prions that have been discovered in natural *S. cerevisiae* isolates.

7) Reviewer 2 also pointed out some issues with discussion, which should be considered to improve the clarify of the paper.

We thank Reviewer 2 for pointing out that our discussion missed an opportunity to integrate our various results, particularly effects on protein synthesis. We have significantly revised our discussion to address this comment.

Reviewer #1:In the manuscript titled "A prion accelerates proliferation at 1 the expense of lifespan", authors Garcia et al., reported that pseudouridine synthase Pus4 acts as a prion switch that regulate the epigenetic state of cell growth by promoting protein synthesis. Overall, this is quite an intriguing paper to read. Most of the experiments were well thought out and designed, most of the data presented are clear and support their conclusions. However, there are a few issues that should be further addressed or clarified before it is accepted for publication.1. In Figure 4G, it does appear that the cytoplasmic localization of Pus4 in [BIG+] cells is ER, maybe rough ER, which may support its function in mRNA pseudouridinylation and promoting protein synthesis. This may be further verified by using inducing [BIG+] in GFP-tagged ER marker or rough ER marker strains.

We thank the reviewer for the suggestion that we determine whether Pus4 protein could be co-localized with the endoplasmic reticulum (ER). We include Author response image 2 showing fusions to two fluorescent proteins: (1) GFP-Pus4 as used in Figure 4G, and (2) dsRed-HDEL, containing a C-terminal sequence that promotes retention of the dsRed protein in the ER. Since we do not see significant overlap in the signal from the two proteins, we conclude that these data do not support Pus4 protein being localized specifically to the ER. Understanding what structures they might co-localize with will be an exciting area for future study. We defer to the editor's judgment on whether these micrographs should be included in the Supplementary Material.

2. The data in Figure 5B are the least clear and convincing in this paper. It is difficult to tell the difference between [BIG+] and pus4del. These data should be quantified in the same way as Figure 7.

Thank you for the suggestion. This point was incorporated into Essential Revisions request #3; please see our response above.

3. The assessment of RNA levels in [BIG+] cells is the biggest issue in this manuscript. RNA-seq from bulk cell sample should be used with spike-in controls normalizes to cell count in order to detect global mRNA level changes (PMID: 23101621). Also, it has been shown clearly that cell size is positively correlated with RNA content (PMID: 30718850). Hence, the levels of RNA or mRNA in each cell should be carefully determined, either by bulk RNA-seq with spike-in control, or single cell RNA-seq.

Thank you for the suggestion. This point was incorporated into Essential Revisions request #4; please see our response above.

4. Figure 6 shows that [BIG+] cell growth was not efficiently inhibited by rapamycin and that [BIG+] cells have higher levels of polysomes. Hence, it should be tested whether TOR signaling is hyperactivated by [BIG+] and that whether this hyperactivation can be efficiently inhibited or attenuated by rapamycin. This can be done by analyzing the phosphorylation levels of Sch9, a target of TOR.

Thank you for the suggestion. This point was incorporated into Essential Revisions request #5, therefore please see our response above.

Reviewer #2:The manuscript by Garcia et al., demonstrates a unique case of post-transcriptional modification "pseudouridylation", on growth adaptation and its effect on lifespan. This modification is induced by transient overexpression of the highly conserved pseudouridine synthase PUS4/TRUB. Authors demonstrated that prion like state of the PUS4 gene affects yeast cell physiology and phenotypes including, growth rate, cell size and lifespan. Authors also connected these phenotype changes to translational regulation of small number of proteins with rare codons. Interestingly these phenotypes are altered without changing mRNA level, including total tRNA and ribosomal RNAs. In addition, global level of protein translation and/or abundance was not affected by this process. They concluded these observations in the light of several carefully designed and elegant experiments. Overall, findings are interesting, and manuscript is written well. As such, the manuscript will greatly promote progress in the field of studying the molecular and cellular mechanisms of aging. However, there are several points to be clarified/improved, which will further strengthen the study. Therefore, the manuscript is suitable for eLife, and I recommend its publication after the following revisions are addressed.Major Comments:1 – The abstract and introduction lack deepness and incorporate some unclear concepts. For example:Line 16 – The opening sentences of the abstract introduce a jargon that is not true for organismal aging. That can be true in the single cell level for some cell types and for some organisms, however; there is no general concept in the aging field that organisms are evolved to select one of these strategies. This sentence was not clarified in the introduction part with further details on the origin of this statement along with necessary references.Line 18: "Lifespan is often inversely correlated with cell size and proliferation". Similarly, this argument in the abstract was not explained in the introduction. Although, there are references regarding to the inverse correlation of cell size and lifespan, there has not been much argument/discussion and related reference about inverse correlation of proliferation rate and lifespan. In fact, in yeast, there is no defined general concept that describes inverse correlation between proliferation rate and aging. More clarification is needed on this statement.

This point was written as Essential Revisions request #1. Please see our response above.

2 – Line 727: Authors claim "These observations suggest that epigenetic, and potentially prion-mediated control of mechanisms like the [BIG+] prion that we have described here, may be widespread in nature". Out of 23 wild yeast isolates, authors only tested four of them and showed the cell size differences. Out of these 4 isolates, only 2 of them showed decreased translation. Authors are clearly cherry picking the results to convince that the observed BIG phenotype is widespread. From the data they present, it is not true. These wild isolates have been previously characterized to have many SNPs and different phenotypes under different stress conditions. Transient inhibition of Hsp70 may cause differences in cell phenotypes, including cell size by acting on different mechanism. The direct testing of BIG phenotype should be performed through transient expression of PUS4 in these strains and measuring the cell size. Also, strains with increased cell size should be subjected to RLS or CLS measurement to align with authors' conclusion. In addition, YPD medium specificity of BIG phenotype already indicates that it is not widespread and it is condition specific. I do not think any of these wild isolates' niches were similar to the YPD condition in their natural environment. Overall, this part is very weak and with the current version, it does not add any further strength to the findings.

Thank you for the helpful comments. This point was incorporated into Essential Revisions request #6. Please see our response above.

3 – Authors conclude that this newly identified prion-based mechanism engages an altered translational program to favor a 'live fast, die young' strategy. There are several interesting findings in this manuscript, however they are somehow disconnected from each other. Basically how these findings are complementing to each other and to their role in observed phenotype need deeper discussion. In the discussion part, authors are only focusing on altered modification of TEF1/TEF2 elongation factors. Some of their findings have not been mentioned. For example, findings of 130 differentially expressed proteins with decreased codon adaptation index along with the findings of no decrease in global protein level are very interesting results. However, the authors seem to avoid discussing these findings. In addition, their findings about [BIG+] cells enhancing translation of some proteins, especially those containing a greater frequency of rare codons need more highlight and to be connected to other findings.

Thank you for the helpful comments. This point was incorporated into Essential Revisions request #7. Please see our response above.

Reviewer #3:The authors report the discovery and characterization of [BIG+] a new yeast prion which shares some properties with other well-studied yeast prions, but also shows some differences, for example in chaperone dependency and physical state. That this prion can apparently control cell growth behaviour of both lab and wild strains of the organism is intriguing, but the precise mechanism by which this is achieved is not fully established in this study. In ruling out any impact on transcription and steady state mRNA/tRNA levels the focus is on a potential post-transcriptional mechanism involving differential synthesis of a relatively large number of proteins. Among the proteins identified there are a number of candidates that could explain the observed [BIG+] phenotypes of cell size and life span changes, but we are left with a list without any one in particular being pinpointed or directly linked to the [BIG+] phenotype.There are two issues where I have some concerns:1. The authors provide several lines of evidence that [BIG+] impacts on the translation of a subset of mRNAs, but I felt the analysis (cf Figure 6) was of relatively low resolution or indirect. Using a coomassie stain of a 1D SDS-PAGE (Fig6G) analysis is a very low resolution approach to look for global differences in protein levels and it would be relatively straightforward – and more informative – if a high res analysis was undertaken. For example, a 2D-SDS PAGE study using pulse-35S-Met-labelled proteins would identify any major changes in the rates of synthesis of at least the higher abundance proteins in [BIG+] vs [big-] cells and give strong evidence for the proposed altered translational programme. The approach they do use – although a clever one – uses quantitation of GFP fluorescence using an N terminally located GFP for 4000+ different proteins. Ideally one would have liked to have seen some evidence that those proteins identified by this screen also showed differences in levels in their native form i.e. without GFP fusions.

Thank you for the helpful comments. This point was incorporated into Essential Revisions request #5. Please see our response above.

Also, the shift in ribosome loading shown in Figure 6H; could the increase in the size of polysomes reflect a defect in translation termination?

Higher polysome intensity could indeed result from slower elongation or inefficient termination. We have added more discussion of these data, including addressing this specific point, to the Discussion section (lines 662–685).

In the future, we wish to use ribosome profiling to address this question more thoroughly but decided that this type of analysis was beyond the scope of the current manuscript.

2. From the data shown it is not clear what the physical form of the Pus4 protein is, in [BIG+] cells. In a relatively low resolution image of Pus4-GFP fusions (Figure 4G) the authors argue that there is a "substantial fluorescence in a fragmented network throughout the cytoplasm" only in [BIG+] cells but to make this statement we need to see higher res images. I was not convinced by the images shown. If they do exist they are clearly not the only form seen in [BIG+] cells as in Figure 8C a wild strain shows Pus4-GFP puncta. Why this difference? Furthermore, the majority of Pus4 in the [BIG+] lab strain is located in the nucleolus; how do we know this is the nucleolus and what happens to the contents of the nucleolus in the cytoduction experiments reported here? A more biochemical approach into analysis of the state of the Pus4 protein is also needed using the methods applied to the study of other yeast prions?

Thank you for this feedback. Please see how we addressed general issues with GFP signal in our response to Essential Revisions request #2. This also allowed us to improve our images of the wild strain shown in Figure 8C.

To our knowledge, the contents of the nucleolus are not exchanged during a cytoduction experiment, since there is no nuclear fusion (Vallen et al., 1992, PMID 1607389). Because we have observed that some Pus4 protein also resides in the cytoplasm in [*BIG*^+^] cells, we presume that at least some of the prionogenic form of Pus4 is also located here, permitting transmission from exchange in this compartment alone.

We agree that more extensive biochemical analysis of Pus4 in the context of its prionogenic conformation warrants further investigation, but believe that these types of experiments are beyond of the scope of the current study.